# Softly Constrained Denoisers for Diffusion Models

## Abstract

Diffusion models struggle to produce samples that respect constraints, a common requirement in scientific applications. Recent approaches have introduced regularization terms in the loss or guidance methods during sampling to enforce such constraints, but they bias the generative model away from the true data distribution. This is a problem, especially when the constraint is misspecified, a common issue when formulating constraints on scientific data. In this paper, instead of changing the loss or the sampling loop, we integrate a guidance-inspired adjustment into the denoiser itself, giving it a soft inductive bias towards constraint-compliant samples. We show that these *softly constrained denoisers* exploit constraint knowledge to improve compliance over standard denoisers, and maintain enough flexibility to deviate from it when there is misspecification with observed data.

## 1 Introduction

Generating realistic data that satisfies specific constraints is a fundamental requirement across numerous applications in scientific discovery, ranging from finding solutions for ODEs (Chen et al., 2018) to designing proteins with certain properties (Gruver et al., 2023). Deep learning techniques have been proposed to solve many of these problems, with varying degrees of success (Chuang & Barba, 2022; Cho et al., 2022; Rezaei et al., 2022; McGreivy & Hakim, 2024). One of the most popular frameworks used in differential equation-based applications is that of *Physics-Informed Neural Networks* (Raissi et al., 2019), where the differential equation residual is used to "inform" the training objective of the neural network, typically through the addition of a residual that quantifies how much the neural network solution deviates from "obeying" the differential equation.

A common pain point in these applications of deep learning has been that neural networks struggle to balance the competing objectives of maintaining data fidelity while satisfying the constraints. Without careful fine-tuning, these methods tend to get stuck on poor local minima that do not reflect the true data distribution (Krishnapriyan et al., 2021) or simply result in solutions that do not fulfill the constraints (Karnakov et al., 2024). This is particularly problematic when the training data deviates from the mathematical model used to formulate the constraints (Finzi et al., 2021; Zou et al., 2024).

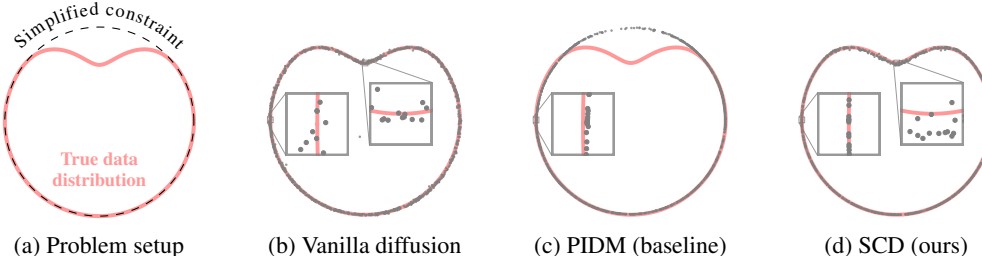

| (a) Problem setup | (b) Vanilla diffusion | (c) PIDM (baseline) | (d) SCD (ours) |

Figure 1: Using a regularizer-based Physics Informed Diffusion Model to enforce the misspecified constraint makes it hard to capture the true data distribution compared to an unconstrained vanilla diffusion. Our method makes the best of the constraint information and vanilla diffusion properties, where it satisfies the partial constraint when it is useful, while maintaining enough flexibility to capture the true geometry.

Diffusion models (Ho et al., 2020; Song et al., 2021) have garnered attention due to their high sample quality, with recent methods exploring their use in inverse problems (Chung et al., 2023a; Song et al., 2023a; Boys et al., 2024) and solving physical differential equations (Jacobsen et al., 2025; Bastek et al., 2025). Some of these diffusion-based approaches have explored the use of pre-trained unconditional models, i.e., models that learn the Stein score, to learn a "guidance" adjustment term on some constraint to guide the diffusion process towards data points that satisfy it (Chung et al., 2023a; Song et al., 2023a). Another branch of research has explored the use of optimization during training the model using a regularizer while training the model (Jacobsen et al., 2025; Bastek et al., 2025). In the former approach, approximations in the inference-time adjustments cause bias in the modeled distribution, and the end result may not improve constraint satisfaction significantly over a standard diffusion model, as they are based on various simplifying approximations. The latter approaches can make optimization more difficult and bias the generative distribution away from the data (Zou et al., 2024; Baldan et al., 2025), as illustrated in Fig. 1.

In this paper, we present *Softly Constrained Denoisers* (SCD) for diffusion models. Inspired by guidance literature (Chung et al., 2023a; Song et al., 2023a;b), we propose a method to endow diffusion model denoisers with constraint knowledge by incorporating a differentiable residual function into the forward pass of the network. Using this approach, we considerably improve constraint satisfaction compared to vanilla diffusion models and approximate guidance-based methods, while keeping a low amount of distributional bias. We present our main ideas and results on an illustrative example and a representative PDE problem.

The contributions of this paper are summarized as follows.

- We propose a method to transform any denoiser architecture to a "softly constrained denoiser" that has an inductive bias towards generating samples that satisfy the constraint. The constraints are embedded in the neural net architecture through the addition of a guidance-like adjustment term that is optimized end-to-end. This adds little computational overhead and preserves the asymptotic data fitting guarantees of standard diffusion models.

- We prove that previous regularizer-based methods cause bias that breaks the standard distribution-modeling guarantees of diffusion models. We show empirically that these methods perform especially poorly with constraint misspecification in representative differential equation problems. In contrast, our method retains all the distribution-modeling guarantees of standard diffusion due to only changing the neural net architecture, and can can extract useful knowledge from the constraint while keeping enough flexibility to deviate from the constraint if the data is not described by it (see Fig. 1 for an example).

- We demonstrate strong empirical results of the effectiveness of our approach through experiments on both illustrative problems and a PDE benchmark, showing that softly constrained denoisers achieve superior constraint satisfaction compared to existing methods, while maintaining sample quality and robustness under constraint misspecification.

## 2 BACKGROUND

Diffusion models generate samples from a data distribution $p(\boldsymbol{x}_0)$ by learning how to denoise samples from a forward *noising* process (Sohl-Dickstein et al., 2015; Ho et al., 2020; Song et al., 2021; Karras et al., 2022), which is generally assumed to be of the form:

$$p(\boldsymbol{x}_t) = \int \mathcal{N}\left(\boldsymbol{x}_t; \boldsymbol{x}_0, \sigma(t)^2 \mathbf{I}\right) p(\boldsymbol{x}_0) \, \mathrm{d}\boldsymbol{x}_0. \tag{1}$$

In simpler terms, clean samples $\boldsymbol{x}_0$ from the data distribution $p(\boldsymbol{x}_0) = p_{\text{data}}(\boldsymbol{x}_0)$ are corrupted by a Gaussian process $\mathcal{N}(\mathbf{0}, \sigma(t)^2 \mathbf{I})$ at time $t$. The corresponding reverse *denoising* process can be formulated as a probability flow ODE (Karras et al., 2022):

$$\mathrm{d}\boldsymbol{x}_t = -\dot{\sigma}(t)\sigma(t)\nabla_{\boldsymbol{x}} \log p(\boldsymbol{x}_t) \, \mathrm{d}t. \tag{2}$$

Starting with a sample from an isotropic Gaussian $\mathcal{N}(\boldsymbol{x}_t; \boldsymbol{x}_0, \sigma_{\max}^2 \mathbf{I})$ and integrating the ODE backwards in time, it is possible to recover a sample from the original data distribution $\boldsymbol{x} \sim p(\boldsymbol{x}_0)$, as long as the score is learned accurately and $\sigma_{\max}$ is large enough (Song et al., 2021). To get $\nabla_{\boldsymbol{x}_t} \log p(\boldsymbol{x}_t)$, we first learn a denoiser conditional on the noise level $t$:

$$\mathcal{L}(\theta) = \mathbb{E}_{t\sim p(t), \boldsymbol{x}_0 \sim p_{\text{data}}, \boldsymbol{x}_t \sim p(\boldsymbol{x}_t \mid \boldsymbol{x}_0)} \left[ w(t) \| D_\theta(\boldsymbol{x}_t, t) - \boldsymbol{x}_0 \|^2 \right], \tag{3}$$

where $w(t)$ and $p(t)$ define the weighting and sampling frequency of noise levels during training, and $D_\theta$ is diffusion model's denoiser with parameters $\theta$. At convergence, $D_\theta(\boldsymbol{x}_t, t) \approx \mathbb{E}[\boldsymbol{x}_0 \,|\, \boldsymbol{x}_t]$. Combined with Tweedie's formula $\mathbb{E}[\boldsymbol{x}_0 \,|\, \boldsymbol{x}_t] = \boldsymbol{x}_t + \sigma(t)^2 \nabla_{\boldsymbol{x}_t} \log p(\boldsymbol{x}_t)$, this ensures that we can recover an approximation of the score $\nabla_{\boldsymbol{x}_t} \log p(\boldsymbol{x}_t)$ with $s_\theta(\boldsymbol{x}_t) = \frac{D_\theta(\boldsymbol{x}_t) - \boldsymbol{x}_t}{\sigma(t)^2}$. Accordingly, the loss in Eq. (3) is also called the *denoising score matching* loss (Vincent, 2011; Song et al., 2021) in the diffusion literature.

**Distributional Bias** The core problem in generative modeling is to learn a surrogate distribution $p_\theta(\boldsymbol{x})$ parameterized by $\theta$ to approximate a data distribution $p_{\text{data}}(\boldsymbol{x})$ (Tomczak, 2022). We call a generative framework *biased* if $p_\theta(\boldsymbol{x})$ does not converge to $p_{\text{data}}(\boldsymbol{x})$ under optimal conditions, i.e., after finding the global optimum of the loss with infinite data, the sampling procedure does not result in samples from $p_{\text{data}}(\boldsymbol{x})$. Importantly, the diffusion model training and sampling in Eq. (3) and Eq. (2) is unbiased in this sense and can thus approximate any data distribution.

**Guided Generation** Assume we have a *constraint function* $c(\boldsymbol{x})$ where the constraint is satisfied when $c(\boldsymbol{x}) = 1$ and not satisfied when $c(\boldsymbol{x}) = 0$. It could be a hard constraint such that $c(\boldsymbol{x}) \in \{0, 1\}$, or a relaxed continuous constraint $c(\boldsymbol{x}) \in [0, 1]$. Given a diffusion model with the output distribution $p(\boldsymbol{x}_0)$, we can turn it into a constrained model with distribution $p(\boldsymbol{x}_0)c(\boldsymbol{x}_0)$ by adjusting the score as follows (Song et al., 2023a; Chung et al., 2023a):

$$s_{\text{adjusted}}(\boldsymbol{x}_t) = \nabla_{\boldsymbol{x}_t} \log p(\boldsymbol{x}_t) + \nabla_{\boldsymbol{x}_t} \log \int c(\boldsymbol{x}_0) p(\boldsymbol{x}_0 \,|\, \boldsymbol{x}_t) \, \mathrm{d}\boldsymbol{x}_0. \tag{4}$$

Many methods have been proposed in the diffusion guidance literature for approximating the second term on the right hand side (Ho et al., 2022; Song et al., 2023a; Chung et al., 2023a; Song et al., 2023b; Rissanen et al., 2025), which generally can be summarized in choosing an approximation for the distribution $p(\boldsymbol{x}_0 \,|\, \boldsymbol{x}_t)$ and an approximation scheme for the integral, e.g. Monte Carlo integration. The idea is to do these approximations at inference time without any changes to the weights of the trained model. While inference-time adjustments are convenient, any error in the approximation results in bias in the output distribution of the diffusion model, and constraint satisfaction is only approximate.

**Regularization** Another approach to constrain the generative space of a model is to use *regularizers* on the training objective. Popularized by *Physics Informed Neural Networks* (Raissi et al., 2019), the general idea is to have an optimization target that is expanded with a differentiable constraint, typically a residual $\mathcal{R} : \mathbb{R}^d \to \mathbb{R}$ related to a differential equation for a physical system, as follows:

$$\mathcal{L}_{\text{target}}(\theta) = \mathcal{L}(\theta) + \lambda \|\mathcal{R}(\theta)\|, \tag{5}$$

where $\lambda \geq 0$ is a hyperparameter that defines how much weight to give to the constraint compliance, the residual $\mathcal{R}$ is used to evaluate the output of a neural network with parameters $\theta$ and $\|\cdot\|$ is some scalar norm of choice, e.g., $L_p$. Although intuitively the target distribution of the learning task should naturally learn to minimize this residual, these PDE-based regularizers can make the loss landscape hard to optimize (Krishnapriyan et al., 2021; Rathore et al., 2024). Further, the optimum of Eq. (5) forfeits the property that $D_\theta(\boldsymbol{x}_t, t) \approx \mathbb{E}[\boldsymbol{x}_0 \,|\, \boldsymbol{x}_t]$ at convergence and the connection between the denoiser and $\nabla_{\boldsymbol{x}_t} \log p(\boldsymbol{x}_t)$ is lost (see Proposition 3.1). Thus, the addition of these targets in the loss function causes bias in the generative distribution (Bastek et al., 2025; Baldan et al., 2025).

**Constraint Misspecification** In science, we build simplified mathematical models around complex phenomena and systems to allow us to study and make use of them. However, since these models are inherently approximations to real phenomena, they are bound to have varying degrees of *misspecification*. We say a constraint is misspecified if it rules out anything that belongs to the true data distribution. Formally, for a constraint $c(\boldsymbol{x}) \in \{0, 1\}$ where $\{\boldsymbol{x} : c(\boldsymbol{x}) = 1\}$ is the constraint-satisfying set, the constraint is misspecified if for any $\boldsymbol{x}$, $p_{\text{data}}(\boldsymbol{x})c(\boldsymbol{x}) \neq p_{\text{data}}(\boldsymbol{x})$. As an example, consider the situation where we model the data with a PDE with specific parameters: The functional form of the PDE could be correct for the data, while the parameters could be incorrect (parametric misspecification, potentially due to measurement errors). On the other hand, the PDE itself could be a simplification (structural misspecification).

For our purposes, constraint misspecification is an issue especially when the underlying generative framework utilizing the constraint is prone to bias, i.e., it does not have guarantees for converging to the data distribution $p_{\text{data}}$ in some limit. An incorrect constraint could push the model even further from $p_{\text{data}}$, amplifying the problem.

## 3 METHODS

Our goal is to propose a method that *(i)* avoids the theoretical bias introduced by regularizers and approximate guidance methods, while *(ii)* still improving the constraint compliance compared to a standard diffusion model. A key limitation of guidance and regularization methods is that they forfeit the theoretical guarantees provided by training a denoiser with Eq. (3) and directly using it for sampling through Eq. (2). One aspect not determined by the diffusion model's mathematics, however, is the neural architecture of the denoiser $D_\theta(\boldsymbol{x}_t, t)$. In principle, this can be parameterized freely. This flexibility motivates us to embed the constraint function $c(\boldsymbol{x})$ directly into $D_\theta(\boldsymbol{x}_t, t)$, imparting an inductive bias towards constraint satisfaction while still allowing deviations when the constraint does not faithfully represent the training data. In this section, we present a principled way of constructing such denoiser parameterizations, building on connections to the guidance literature and Eq. (4).

**Class of Constraints** In this paper, we consider constraints that are possible to write in a form $\mathcal{R}(\boldsymbol{x}) = 0$, where $\mathcal{R}$ is some function. Thus, $c(\boldsymbol{x}) = 1$ for all $\boldsymbol{x}$ in $\{\boldsymbol{x} : \mathcal{R}(\boldsymbol{x}) = 0\}$ and $0$ elsewhere. This could, for instance, be a finite difference based residual formed from a PDE, or a circle in 2D, where $\mathcal{R}(\boldsymbol{x}) = x_1^2 + x_2^2 - 1$. We further assume $\mathcal{R}$ to be continuously differentiable in the input $\boldsymbol{x}$, such that we can calculate gradients $\nabla_{\boldsymbol{x}}\mathcal{R}(\boldsymbol{x})$. We then define a *relaxed constraint function* $l_c(\boldsymbol{x}) = \exp(-\|\mathcal{R}(\cdot)\|)$ for some choice of norm $\|\cdot\|$, similar to the recipe prescribed by Song et al. (2023b), connecting our method with the guidance literature. Importantly, the norm and more broadly the design of $l_c(\boldsymbol{x})$ is a design choice for our method, and we could design it in any way that provides information about the constraint to the denoiser architecture.

**Guidance Approximation** Following common ideas in the guidance literature, we first propose a practical approximation to Eq. (4).This leads to a formula that nudges the denoiser towards satisfying the constraint by using the gradient of a (relaxed) constraint function $\nabla_{\boldsymbol{x}_0} l_c(\boldsymbol{x}_0)$. We then use this approximate form as an inspiration for a denoiser parameterization that learns to calibrate this gradient-based nudge, freeing the base neural network from having to output values that are perfectly aligned with the constraints.

First we choose an approximation to $p(\boldsymbol{x}_0 \,|\, \boldsymbol{x}_t)$ in Eq. (4). A common choice is $p(\boldsymbol{x}_0 \,|\, \boldsymbol{x}_t) = \mathcal{N}(D_\theta(\boldsymbol{x}_t, t), \Sigma_{0|t}^2 \mathbf{I})$, where $\Sigma_{0|t}^2$ is a hyperparameter (Ho et al., 2022; Song et al., 2023a; Chung et al., 2023a; Boys et al., 2024; Peng et al., 2024; Rissanen et al., 2025). Similarly to Chung et al. (2023a), we choose $\sigma_{0|t}^2 = 0$. Thus, $p(\boldsymbol{x}_0 \,|\, \boldsymbol{x}_t)$ turns into a Dirac delta on $D_\theta(\boldsymbol{x}_t, t)$ and taking the integral in Eq. (4) with this choice produces the following:

$$s_\theta^{\text{guided}}(\boldsymbol{x}_t) \approx \frac{D_\theta(\boldsymbol{x}_t, t) - \boldsymbol{x}_t}{\sigma_t^2} + \nabla_{\boldsymbol{x}_t} \log l_c\left(D_\theta(\boldsymbol{x}_t, t)\right). \tag{6}$$

This expression results in a vector-Jacobian product through the denoiser, which can result in considerable computational overhead:

$$\nabla_{\boldsymbol{x}_t} \log l_c\left(D_\theta(\boldsymbol{x}_t, t)\right)^\top = \nabla_{D_\theta} \log l_c\left(D_\theta(\boldsymbol{x}_t, t)\right)^\top \nabla_{\boldsymbol{x}_t} D_\theta(\boldsymbol{x}_t, t). \tag{7}$$

To obtain a more efficient approximation, we recall that the Jacobian is connected to the denoising covariance through $\nabla_{\boldsymbol{x}_t} D_\theta(\boldsymbol{x}_t, t) = \frac{\text{Cov}[\boldsymbol{x}_0 \,|\, \boldsymbol{x}_t]}{\sigma(t)^2}$ (Boys et al., 2024). Analogous to our earlier treatment of $p(\boldsymbol{x}_0 \,|\, \boldsymbol{x}_t)$, we approximate the conditional covariance as $\text{Cov}[\boldsymbol{x}_0 \,|\, \boldsymbol{x}_t] \approx \boldsymbol{\Lambda}_t$, where $\boldsymbol{\Lambda}_t$ is some diagonal matrix. Altogether, this yields the following denoiser correction:

$$D_{\theta,\text{guided}}(\boldsymbol{x}_t, t) = D_\theta(\boldsymbol{x}_t, t) + \boldsymbol{\Lambda}_t \sigma(t)^2 \nabla_{\boldsymbol{D}_\theta} \log l_c(D_\theta(\boldsymbol{x}_t, t)). \tag{8}$$

**Softly Constrained Denoiser** The key idea in our paper is as follows: since Eq. (8) tends to generate samples that satisfy the constraint for a suitably chosen $\boldsymbol{\Lambda}_t$, we can form a denoiser parameterization that uses the same structure for easier constraint compliance through a learned $\boldsymbol{\Lambda}_t$ term

$$D_\theta(\boldsymbol{x}_t, t) = D_\theta^{\text{orig}}(\boldsymbol{x}_t, t) + \boldsymbol{\gamma}_\theta(\boldsymbol{x}_t, t)\sigma(t)^2 \nabla_{\boldsymbol{D}_\theta^{\text{orig}}} \log l_c(D_\theta^{\text{orig}}(\boldsymbol{x}_t, t)), \tag{9}$$

where $D_\theta^{\text{orig}}(\boldsymbol{x}_t, t)$ is the original denoiser output and $\boldsymbol{\gamma}_\theta(\boldsymbol{x}_t, t)$ is a learnable scaling factor that can be parameterized by the same base neural network as $D_\theta^{\text{orig}}(\boldsymbol{x}_t, t)$, or by a separate network. The model can then be trained using the standard denoising score matching loss in Eq. (3). Crucially, the correction term in Eq. (9) only evaluates the gradient of the constraint $l_c$ until $D_\theta$, avoiding the

costly calculation of a vector-Jacobian product in the forward pass. While we adopt this particular approximation, many alternative (and potentially more sophisticated) formulations of the guidance formula are possible. Each such choice would naturally define a corresponding softly constrained denoiser, making our approach a general recipe for deriving new variants. In our experiments, we focus on Eq. (9) and leave more sophisticated parameterizations for future work. We show the training algorithm explicitly in Algorithm 1 and visualise the new denoiser architecture in Fig. 2.

---

**Algorithm 1** Loss for SCD

---

**Require:** Data $p_{data}(\boldsymbol{x})$, constraint $l_c(\boldsymbol{x})$, noise $\sigma(t)$, weights $w(t)$, noise $p(t)$
1: Init $\boldsymbol{\theta}$; Sample $\boldsymbol{x}_0, t, \boldsymbol{\epsilon}$
2: $\boldsymbol{x}_t \leftarrow \boldsymbol{x}_0 + \sigma(t)\boldsymbol{\epsilon}$
3: **Compute SCD Output $\hat{x}_{\boldsymbol{\theta}}$:**
4: $\hat{\boldsymbol{x}}_{base} \leftarrow D_{\boldsymbol{\theta}}^{orig}(\boldsymbol{x}_t, t)$
5: $g \leftarrow \nabla_{\boldsymbol{x}} \log l_c(\hat{\boldsymbol{x}}_{base})$
6: $\hat{\lambda} \leftarrow \gamma_{\boldsymbol{\theta}}(\boldsymbol{x}_t, t)\sigma(t)^2$
7: $\hat{\boldsymbol{x}}_{\boldsymbol{\theta}} \leftarrow \hat{\boldsymbol{x}}_{base} + \hat{\lambda} \cdot g$
8: **Output Loss:**
9: $\mathcal{L} \leftarrow w(t)\|\hat{\boldsymbol{x}}_{\boldsymbol{\theta}}(\boldsymbol{x}_t, t) - \boldsymbol{x}_0\|^2$

---

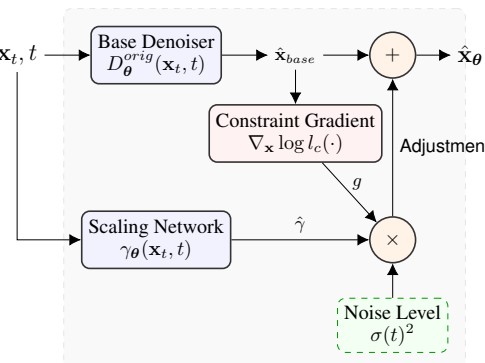

Figure 2: Architecture of SCD.

## 3.1 ANALYSIS OF REGULARIZATION BIAS AND ELBO DEGRADATION

In this section, we formally analyze the impact of introducing a regularization term to the training objective, as proposed in PIDM (Bastek et al., 2025). We demonstrate that while this regularization enforces constraint satisfaction, it biases the denoiser away from the true conditional expectation and also necessarily degrades the variational lower bound (ELBO) of the diffusion model (Ho et al., 2020; Kingma et al., 2021).

Consider the regularized objective function for a specific noise level $t$ (ignoring the weighting term $w(t)$ for brevity):

$$\mathcal{L}_{\text{reg}}(\theta) = \mathbb{E}_{x_0, x_t}\left[\|D_\theta(x_t, t) - x_0\|^2\right] + \lambda\|\mathcal{R}(D_\theta(x_t, t))\|^2, \tag{10}$$

where $\lambda > 0$ is the regularization weight and $\mathcal{R}$ is the constraint residual.

**Proposition 3.1.** *Let $D_{reg}^*(x_t, t)$ be the denoiser that minimizes the regularized objective $\mathcal{L}_{reg}$. The optimal denoiser output is shifted from the true conditional mean of the data distribution by a term proportional to the gradient of the residual. Specifically:*

$$D_{reg}^*(x_t, t) = \mathbb{E}[x_0|x_t] - \lambda \left[\nabla_y \mathcal{R}(y)\right]^\top \mathcal{R}(y)\Big|_{y = D_{reg}^*}, \tag{11}$$

*where we assume a scalar constraint residual for simplicity, or $\mathcal{R}$ represents the norm function directly. Since $\mathbb{E}[x_0 \,|\, x_t]$ does not generally satisfy the constraint, and the residual or its gradient are zero only when the constraint is satisfied, the optimal denoiser output is shifted. The asymptotic distributional guarantees of diffusion models are thus lost since there is no connection between the optimal denoiser and the score function $\nabla_{\boldsymbol{x}_t} \log p(\boldsymbol{x}_t)$.*

**Proposition 3.2.** *The use of the regularized denoiser $D_{reg}^*$ strictly increases the Evidence Lower Bound (ELBO) loss component compared to the vanilla denoiser $D_{vanilla}^*$. That is, the model's approximation of the data likelihood deteriorates.*

See Appendix A for the proofs. Note that the PIDM paper (Bastek et al., 2025) also considered regularising based on a DDIM integration output, but their best results were obtained by regularising the $D_\theta(x_t, t)$ directly.

## 4 RELATED WORK

**Diffusion Models Applied to PDEs**   Jacobsen et al. (2025) propose conditional PDE generation using a Controlnet-like conditioning structure (Zhang et al., 2023) and an inference-time adjustment where the final samples are optimized to have a small PDE residual. Bastek et al. (2025) present Physics-Informed Diffusion Models (PIDM), a framework to train DDPM-based diffusion models with a PDE residual as a regularizer term to minimize along the loss function. Several works utilize DPS-like guidance (Chung et al., 2023a) for PDE data assimilation, targeting noisy measurements (Shysheya et al., 2024), constraint satisfaction (Huang et al., 2024), or infinite-dimensional Banach spaces (Yao et al., 2025). Similarly, Cheng et al. (2025) employ projection-based sampling (Zhu et al., 2023). Unlike these, our method avoids approximate inference-time guidance. Furthermore, our parameterization is orthogonal to recent architecture-focused works on neural operators (Hu et al., 2025; Oommen et al., 2024) or GNNs (Valencia et al., 2025), as it remains compatible with any base architecture.

**Injecting Measurement Structure for Training Inverse Problem Solvers**   Mathematically, the closest work is the likelihood-informed Doob's h-transform by Denker et al. (2024), who finetune adapters using observation gradients $\nabla_{\boldsymbol{x}_0} p(\boldsymbol{y} \,|\, \boldsymbol{x}_0)$, similar to our constraint-informed parameter- ization using $\nabla_{\boldsymbol{x}_0} l_c(\boldsymbol{x}_0)$. However, our motivation and methodology differ: they use likelihoods $p(\boldsymbol{y} \,|\, \boldsymbol{x}_0)$ for Bayesian inference from noisy observations, whereas we *define* $l_c(\boldsymbol{x}_0)$ to restrict gener- ation to a constrained subset. Their goal is an alternative to inference-time adjustment, while we seek to alleviate distributional biases and constraint misspecification. Furthermore, we train from scratch, whereas they finetune adapters on larger models. Finally, Liu et al. (2023b); Chung et al. (2023b) propose embedding structure via bridge processes that interpolate between condition and target; this is inapplicable to our setting, as we do not target a translation problem.

**Inference Time Adjustment for Inverse Problems**   Many methods target adjusting diffusion models at inference time to solve inverse problems, many of them formally targeting approximation of Eq. (4). Song et al. (2021) was one of earliest papers to propose inference-time adjustments to the diffusion model to solve problems like inpainting and color restoration. Chung et al. (2022); Wang et al. (2023a); Zhu et al. (2023) propose more advanced methods for a wider range of inverse problems. The first methods to make the explicit connection to Eq. (4) were (Ho et al., 2022; Chung et al., 2023a; Song et al., 2023a). While the method by Song et al. (2023a) only worked for linear inverse problems, Song et al. (2023b) generalized it to general guidance functions through Monte Carlo integration of Eq. (4). Works focused on improving the $p(\boldsymbol{x}_0 \,|\, \boldsymbol{x}_t)$ approximation include (Boys et al., 2024; Peng et al., 2024; Rissanen et al., 2025).

**Hard-Constraint Diffusion via Modified Dynamics**   Several works impose constraints by *changing the diffusion dynamics* so that the support is restricted by construction. Riemannian diffusion models and flow matching models move the noising and denoising processes to a target manifold, enabling sampling on spheres, tori, hyperboloids, and matrix groups but requiring smooth geometry and geo- metric operators De Bortoli et al. (2022); Huang et al. (2022); Chen & Lipman (2024). Fishman et al. (2023a;b); Lou & Ermon (2023) propose to use noising processes that are constrained within convex sets defined by inequality constraints. while (Liu et al., 2023a) tackle the problem by learning standard diffusion models in a dual space created using a mirror map. *Star-shaped DDPMs* tailor noise to exponential-family distributions suited to constrained manifolds Okhotin et al. (2023). These methods provide hard constraints but are typically specialized to particular geometries or constraint classes.

**Optimizing Samples to Match with Constraints**   Ben-Hamu et al. (2024) generate samples with constraints by optimizing a source point in the noisy latent space such that the generative ODE solution matches with the constraint. Tang et al. (2024) instead optimize the noise injected during the stochastic sampling process. Poole et al. (2023) generate samples by directly optimizing the target im- age within a constrained space (e.g., images parameterized by a neural radiance field (Mildenhall et al., 2021)), while minimizing the diffusion loss for the image. Wang et al. (2023b) extends the method with a particle-based variational framework. In a similar manner, Mardani et al. (2024) formulate the sampling process as a optimizing a variational inference distribution on the clean samples.

**Distributional Constraints**   Khalafi et al. (2024) formulate a distributional constrained generation task with the constraint that the generative distribution should have a KL divergence below a threshold to a set of auxiliary distributions $q^i$. Khalafi et al. (2025) extend the idea to compositional generation.

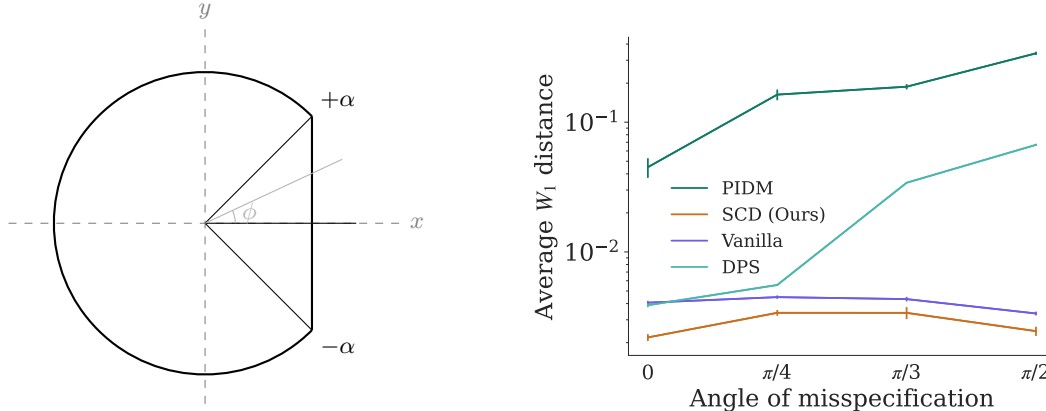

Figure 3: (a) "Chop" misspecification for our experiments. For all points with angles $-\alpha < \phi < \alpha$ the $x$ coordinate is projected to $x = \cos\alpha$. (b) Average Wasserstein-1 distances on the circles examples with varying degrees of misspecification on "Chop". Vanilla and SCD keep steady values of $W_1$ distance which indicates their flexibility to learn the true data distribution, whereas PIDM consistently increases with higher levels of misspecification. Means drawn with two standard deviations.

**Soft Inductive Biases**  Finzi et al. (2021) propose using "dual path" layers to build a neural network, where one path uses a hard-constrained layer, e.g. a rotationally equivariant layer, and the other uses a more "relaxed" layer. By assigning a lower prior probability to the relaxed path, a soft inductive bias towards solutions that satisfy the constraint is imposed, without restricting the possible hypothesis space for the neural network.

## 5  EXPERIMENTS

We first showcase and analyse our method on an illustrative set of toy examples in Section 5.1. In Section 5.2, we evaluate our method on the Darcy flow PDE data set, a commonly used benchmark in the diffusion scientific constrained generation literature (Jacobsen et al., 2025; Bastek et al., 2025). For all experiments, $\gamma_\theta$ was implemented as a small 2-layer MLP that takes $D_\theta(\boldsymbol{x}_t)$ and $t$ as input and outputs a scalar. Details are available in Appendix D. We also use a modified version of the loss function by Karras et al. (2022): in our experiments we observed that the models had issues with refining the fine-grained details at the lower noise levels, and modifying the distribution from which the noise levels are sampled significantly improved the results across all models. The details of the modification are available in Appendix D.1

### 5.1  ILLUSTRATIVE EXAMPLES

We explore our method in new variants of the toy data set introduced by Bastek et al. (2025), introducing new misspecification modalities. The data itself is simply points sampled from the unit circle centered at the origin, and we train a diffusion model to learn to produce samples on the circle. Given a sample $(x, y) \in \mathbb{R}^2$, we define a residual function for this setting with the following equation:

$$\mathcal{R}_{\text{circle}}(x, y) = c \left( \sqrt{x^2 + y^2} - 1 \right)^2, \tag{12}$$

where $c \in \mathbb{R}^+$ is a constant that scales the function. The architecture and training details are shown in Appendix D.2. We use $r_c = \exp(-\mathcal{R}_{\text{circle}}(x, y))$ as our constraint term for these experiments. We then evaluate the performance of vanilla diffusion, regularizer based diffusion (PIDM) and our method on a few examples of misspecification. Specifically, we use Eq. (12) with PIDM and our method on variations of the circle, namely a circle with a "dent" on top and a circle that is "chopped" after a particular $x$ coordinate.

For "Dent", we use a circle with a polynomial interpolation at the top half. This produces the shape seen in Fig. 1, the details on the interpolation are available in Appendix D.2. For "Chop", as illustrated

Table 1: Wasserstein-1 distances measured from the true data distribution with each method. All the values are presented as the mean with two standard deviations across 100 estimates with 1000 samples taken from each method. **Bold** shows the best result. Lower is better. All values are multiplied by $10^{-3}$.

| Method | Unit circle | Dent | Chop ($\alpha = \frac{\pi}{2}$) |
|---|---|---|---|
| Vanilla | $3.91 \pm 0.18$ | $7.53 \pm 0.46$ | $3.34 \pm 0.11$ |
| PIDM | $4.04 \pm 0.20$ | $6.65 \pm 1.44$ | $33.75 \pm 3.37$ |
| SCD (ours) | $\mathbf{2.16 \pm 0.15}$ | $\mathbf{5.6 \pm 0.44}$ | $\mathbf{2.42 \pm 0.16}$ |

in Fig. 3, we define an angle threshold $-\alpha < \phi < \alpha$ for which all points on the circle with angle $\phi$ have their $x$ coordinate projected to $\cos\alpha$.

The results of using vanilla diffusion, PIDM, and our method on the standard circle and the two described misspecifications are summarized in Table 1. We can see that in the case of the unit circle, all methods achieve relatively similar $W_1$ distances, indicating that all are capable of capturing the underlying geometry. For "Dent", we see that PIDM starts to have higher values, and we can see from Fig. 1 that this is because training with PIDM makes the model incapable of putting samples on the dent. This issue is most prominent with "Chop", where the $W_1$ distance with PIDM is almost an order of magnitude higher than the other two methods: this is because it does not put any mass on the projected line, it only learns the samples on the arc along the circle.

To measure the effect of varying the misspecifications on the toy data, we vary the angle for "Chop" and note the change on the $W_1$ distance. This is seen on Fig. 3. Both vanilla diffusion and our method stay relatively steady for all different $\alpha$, while PIDM sees a steady increase with higher degrees of misspecification. Note the $W_1$ distance for PIDM is worse than vanilla diffusion even when the constraint is correctly specified at $\alpha = 0$, potentially due to the distribution being biased *along the circle* even if the constraint is satisfied. Similar issues were visually noticed in Bastek et al. (2025) when using a high weight on the regularization term.

### 5.2 Darcy Flow

The task is to learn how to produce samples of a permeability field $K(x, y)$ and a pressure field $p(x, y)$ in two dimensions $(x, y) \in \mathbb{R}^2$ that satisfy the following differential equation:

$$\mathcal{R}_{\text{Darcy}}(K, p) = \nabla \cdot (K\nabla p) + f = 0, \tag{13}$$

where $f$ is the measurement of the flow of some fluid through a porous medium, and corresponds to the divergence of the vector field defined by $K\nabla p$ pictured in Fig. 4, or the *net* amount of fluid entering and exiting a given point. Particularly, this model is developed to study systems with *laminar* flow, for more general cases there must be additional adjustments Schlichting & Gersten (2017); Potter & Ramadan (2012). To make use of this equation in practice, we must either make assumptions about the material (e.g. an isotropic porosity), take potentially noisy measurements of the fluid's flow and applied pressure field, or both. We classify all these issues as sources of *misspecification* on this setting. For a brief treatment on the topic and its range of applications, see Appendix B.

This differential equation doubles as a residual function we can use to verify our generated solutions. For the experiments, the differential terms are estimated through finite differences using the same stencils used by Bastek et al. (2025), meaning $K$ and $p$ are sampled as matrices in $\mathbb{R}^{n \times n}$. We use this implementation to define our constraint adjustment $r_c = \exp(-|\mathcal{R}_{\text{Darcy}}(K, p)|)$, where $K$ and $p$ are the denoiser outputs. To tackle this problem, we use a diffusion model with a UNet backbone developed by Karras et al. (2023), and use a discretization of $K$ and $p$ in $\mathbb{R}^{64 \times 64}$. The architecture, training details and runtimes are available in Appendix D.3. We highlight the small relative overhead in runtime between our method and a vanilla score matching implementation.

**Distributional Fidelity, Misspecification and Residuals** Darcy Flow is a mathematical model particularly used to infer properties of a material in a real physical system. As such, using this model can be prone to different sources of error, ranging from measurement error (e.g., miscalibrated tools, noisy environments) to using it on a system with critical flow, where the fluid is in a state between laminar and turbulent flow. In this section, we present experiments on a simple case where the measured flow is on a wrong scale (an example of miscalibration).

Table 2: Residual values and validation set NLL of the different methods across different levels of source and sink misspecification. Our method shows good performance across all misspecification levels. Residual values presented are mean absolute residuals across 1000 samples with two standard deviations and NLL values are on the complete validation set, reported in bits/dim. Vanilla does not use the constraint knowledge, so we treat it the same for all levels as a baseline. **Bold** shows the best result, underline shows second best. Lower is better.

| Metric | Method | *Original* $f_{\max} = 10$ | *increasing misspecificication* $f_{\max} = 20$ | $f_{\max} = 30$ | $f_{\max} = 40$ |
|---|---|---|---|---|---|
| Residual | Vanilla | $0.157 \pm 0.071$ | $0.157 \pm 0.071$ | $0.157 \pm 0.071$ | $0.157 \pm 0.071$ |
| | Guided Vanilla | $0.141 \pm 0.063$ | $0.139 \pm 0.067$ | $0.140 \pm 0.065$ | $\underline{0.139 \pm 0.066}$ |
| | PIDM | $\mathbf{0.025 \pm 0.010}$ | $\mathbf{0.332 \pm 0.007}$ | $\underline{0.647 \pm 0.007}$ | $0.963 \pm 0.009$ |
| | SCD (ours) | $\underline{0.113 \pm 0.048}$ | $\underline{0.106 \pm 0.049}$ | $\mathbf{0.114 \pm 0.059}$ | $\mathbf{0.118 \pm 0.051}$ |
| NLL | Vanilla | $\mathbf{-10.5}$ | $\mathbf{-10.5}$ | $\mathbf{-10.5}$ | $\mathbf{-10.5}$ |
| | Guided Vanilla | $\underline{-10.3}$ | $\underline{-10.3}$ | $\underline{-10.3}$ | $\underline{-10.3}$ |
| | PIDM | $-3.7$ | $-3.7$ | $-3.6$ | $-3.2$ |
| | SCD (ours) | $-8.4$ | $-6.9$ | $-5.9$ | $-5.8$ |

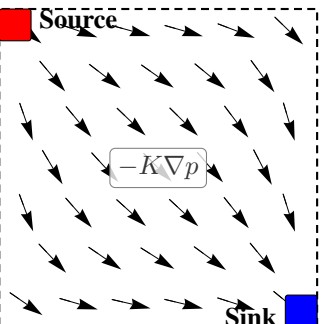

Figure 4: A simplified visual representation of the Darcy velocity field defined by $K\nabla p$.

To induce the misspecification, we test out different values for the measured flow $f$ in the constraint, while keeping the original data. The results using vanilla, guided vanilla, PIDM and our method are shown in Table 2 reported with residual values and NLL. The vanilla diffusion with inference-time guidance is implemented using Eq. (6) following "Loss Guided Diffusion" by Song et al. (2023b), and the guidance scale used for the guided vanilla was tuned to a value of 0.03 based on the residual values with a grid search. Details on this guidance method can be seen in Appendix C. Qualitative visualizations and samples from our method are available in Appendix B.1. We highlight that PIDM is especially sensitive to the induced misspecifications, causing its residuals to go up considerably to the point that it performs worse than a baseline vanilla diffusion model. On the other hand, our model shows relative resilience to these changes. We hypothesize that it can still use the information in the area where the $f$-field is correctly specified as zero, while learning to adapt the gradient information used for the source and the sink. Using a guidance method with vanilla diffusion slightly improves the residuals but not by a significant margin, and the same behavior continues across different misspecification levels.

## 5.3 HELMHOLTZ EQUATION

The Helmholtz Differential Equation is used to model the propagation of waves through (possibly heterogeneous) media. Its 2-dimensional version is given by:

$$\mathcal{R}_{\text{Helmholtz}}(u, a) = \nabla^2 u(\boldsymbol{x}) + k^2 u(\boldsymbol{x}) - a(\boldsymbol{x}) = 0, \qquad \boldsymbol{x} \in (0,1)^2, \tag{14}$$

where $u$ is known as the solution field, $a$ the source field and $k$ the wave number. With this presentation, the solution is given by a spatial wave in 2 dimensions and boundary conditions equal to 0. This equation is particularly important in acoustics and electromagnetics, describing the way sound and light travel through space. It is generally subject to the measurement of $k$, which uniquely defines the solution of the system given the boundary conditions. The wave number is usually estimated with sensor readings in noisy media, which subjects the equation to observation error.

Similar to our approach to Darcy Flow, we use Eq. (14) as the residual to define $r_c = \exp(-|\mathcal{R}_{\text{Helmholtz}(u,a)}|)$, where $u$ and $a$ correspond to the denoiser outputs. Following Yao et al. (2025), we use a discretization of $u$ and $a$ in $\mathbb{R}^{128 \times 128}$ modeled by the Diffusion Model with the same UNet backbone as Darcy and a wave number $k = 1$. The training details are available in Appendix D.4.

Table 3: Residual values and validation set NLL of the different methods across different levels of source and sink misspecification. Our method shows good performance across all misspecification levels. Residual values presented are mean absolute residuals across 1000 samples with two standard deviations and NLL values are on the complete validation set, reported in bits/dim. Vanilla does not use the constraint knowledge, so we treat it the same for all levels as a baseline. **Bold** shows the best result, underline shows second best. Lower is better.

| | | *Original* | *increasing misspecificication* | | |
|---|---|---|---|---|---|
| **Metric** | **Method** | $\sigma_{\text{obs}} = 0$ | $\sigma_{\text{obs}} = 0.05$ | $\sigma_{\text{obs}} = 0.1$ | $\sigma_{\text{obs}} = 0.5$ |
| Residual | Vanilla | $0.035 \pm 0.007$ | $0.035 \pm 0.007$ | $0.035 \pm 0.007$ | $0.035 \pm 0.007$ |
| | FunDPS | $13.84 \pm 3.4$ | $13.86 \pm 3.4$ | $13.84 \pm 3.3$ | $14.2 \pm 3.4$ |
| | DPS | $0.032 \pm 0.004$ | $0.033 \pm 0.005$ | $0.035 \pm 0.004$ | $0.034 \pm 0.003$ |
| | SCD (ours) | $\mathbf{0.025 \pm 0.003}$ | $\mathbf{0.024 \pm 0.003}$ | $\mathbf{0.024 \pm 0.002}$ | $\mathbf{0.023 \pm 0.003}$ |
| NLL | Vanilla | $\mathbf{-21.6}$ | $\mathbf{-21.6}$ | $\mathbf{-21.6}$ | $\mathbf{-21.6}$ |
| | FunDPS | $-7.2$ | $-7.3$ | $-7.2$ | $-7.2$ |
| | DPS | $\mathbf{-21.6}$ | $\underline{-21.5}$ | $\underline{-21.5}$ | $\underline{-21.5}$ |
| | SCD (ours) | $\mathbf{-21.6}$ | $\underline{-21.5}$ | $\underline{-21.5}$ | $\underline{-21.5}$ |

**Distributional Fidelity, Misspecification and Residuals** In this setting, we induce the misspecifications through the addition of Gaussian noise to the wave number. The degree of misspecification is handled by the standard deviation of this noise, and this experiment allows us to measure how sensitive SCD can be to noisy gradients through the constraint function. The results are summarized in Table 3, where we compare our results with the guidance framework FunDPS using a model trained by the authors (Yao et al., 2025), a vanilla diffusion model and DPS guidance on the vanilla diffusion model. The FunDPS sampling is done with a combination of a reconstruction loss based on partial observations and the residual to observe the issues resulting from imbalanced guidance. SCD manages to preserve the NLL with the most competitive residual values in this task. It is evident that using this combined guidance in FunDPS heavily favors the reconstruction error, essentially forfeiting the physical consistency of the generated samples. On the other hand, regular DPS only sees marginal improvements in terms of the residuals compared to the vanilla diffusion model.

## 6 DISCUSSION AND CONCLUSION

In this work, we introduced the Softly Constrained Denoiser (SCD), a simple way to embed constraint knowledge directly into diffusion model denoisers. Unlike guidance-based methods that add constraints only at inference time or regularization-based methods that bias the training distribution, our approach provides a soft inductive bias that improves constraint satisfaction while retaining flexibility when constraints are misspecified. We demonstrated these benefits through illustrative examples (Section 5.1) and PDE benchmarks (Section 5.2), showing that SCD outperforms standard diffusion models and existing constraint-enforcing approaches when the constraint is partially misspecified. Our results suggest that SCD can serve as a drop-in upgrade for diffusion-based generative modeling in applications where constraints are desired.

**Limitations and Future Work** A limitation of our proposed method is that our denoiser form is, at the end of the day, an approximation to the score function for the tilted distribution $r_c(\boldsymbol{x}_0)p(\boldsymbol{x}_0)$. As such, the approximation may limit the usefulness of the inductive bias that it provides. Another limitation is that our model is not guaranteed to generate samples within a particular hard constraint, and if an application requires exact constraint satisfaction, further postprocessing of generated samples is necessary. Finally, even though the inductive bias provided by the denoiser parameterization does not necessarily inhibit the diffusion model from following the data distribution, the parameterization may push the model towards particular biases in practice. Further, in the presence of significant misspecification the new parameterization may cause noise relative to standard denoisers. Future work could explore deriving SCDs based on more advanced guidance approximations and applying them in various scientific application areas. Even better robustness to constraint specification could be achieved by introducing learnable parameters to $l_c(\boldsymbol{x})$ itself.

## REPRODUCIBILITY STATEMENT

To reproduce our results we supply the code for training Darcy Flow and the illustrative examples in a supplementary zip file. We acquire the data for Darcy Flow from Bastek et al. (2025), so we refer reviewers to visit their repository to gather the data set.

Upon acceptance of the paper, we will make a reference code implementation together with the experiments available on GitHub under the MIT license.

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

# APPENDICES

## A PROOFS

**Proposition 3.1.** *Let $D_{reg}^*(x_t, t)$ be the denoiser that minimizes the regularized objective $\mathcal{L}_{reg}$. The optimal denoiser output is shifted from the true conditional mean of the data distribution by a term proportional to the gradient of the residual. Specifically:*

$$D_{reg}^*(x_t, t) = \mathbb{E}[x_0|x_t] - \lambda \left[\nabla_y \mathcal{R}(y)\right]^\top \mathcal{R}(y)\Big|_{y=D_{reg}^*}, \tag{11}$$

*where we assume a scalar constraint residual for simplicity, or $\mathcal{R}$ represents the norm function directly. Since $\mathbb{E}[x_0 \,|\, x_t]$ does not generally satisfy the constraint, and the residual or its gradient are zero only when the constraint is satisfied, the optimal denoiser output is shifted. The asymptotic distributional guarantees of diffusion models are thus lost since there is no connection between the optimal denoiser and the score function $\nabla_{\boldsymbol{x}_t} \log p(\boldsymbol{x}_t)$.*

*Proof.* Let $y = D_\theta(x_t, t)$. The objective can be rewritten as the expected risk for a given input $x_t$:

$$J(y) = \mathbb{E}_{x_0|x_t} \left[\|y - x_0\|^2\right] + \lambda\|\mathcal{R}(y)\|^2. \tag{15}$$

To find the optimum $y^*$, we take the gradient with respect to $y$ and set it to zero:

$$\nabla_y J(y) = \nabla_y \left(\|y\|^2 - 2y^\top \mathbb{E}[x_0|x_t] + \mathbb{E}[\|x_0\|^2]\right) + \lambda \nabla_y \|\mathcal{R}(y)\|^2 \tag{16}$$

$$0 = 2(y - \mathbb{E}[x_0|x_t]) + 2\lambda(\nabla_y \mathcal{R}(y))^\top \mathcal{R}(y). \tag{17}$$

Solving for $y$ yields the result. Consequently, $D_{reg}^*(x_t, t) \neq \mathbb{E}[x_0|x_t]$ unless $\lambda = 0$ or the constraint gradient is zero. $\qquad\square$

**Proposition 3.2.** *The use of the regularized denoiser $D_{reg}^*$ strictly increases the Evidence Lower Bound (ELBO) loss component compared to the vanilla denoiser $D_{vanilla}^*$. That is, the model's approximation of the data likelihood deteriorates.*

*Proof.* The standard diffusion loss $\mathcal{L}_{\text{diff}} = \mathbb{E}[\|D_\theta(x_t, t) - x_0\|^2]$ corresponds to the variational bound on the negative log-likelihood if using a specific weighting $w(t)$(Kingma et al., 2021). It is a well-known result that the unique global minimizer of this quadratic loss is the conditional expectation $D_{\text{vanilla}}^*(x_t, t) = \mathbb{E}[x_0|x_t]$.

Since $D_{reg}^*(x_t, t) \neq \mathbb{E}[x_0|x_t]$ (from Proposition 3.1), and $\mathcal{L}_{\text{diff}}$ is strictly convex with respect to the prediction $y$, it follows that:

$$\mathbb{E}_{x_0|x_t} \left[\|D_{\text{reg}}^*(x_t) - x_0\|^2\right] > \mathbb{E}_{x_0|x_t} \left[\|D_{\text{vanilla}}^*(x_t) - x_0\|^2\right]. \tag{18}$$

Specifically, by the bias-variance decomposition, the increase in the diffusion loss is exactly the squared magnitude of the shift derived in Proposition 3.1:

$$\Delta\mathcal{L} = \|D_{\text{reg}}^*(x_t, t) - \mathbb{E}[x_0|x_t]\|^2. \tag{19}$$

Thus, optimizing the regularized objective $\mathcal{L}_{\text{target}}$ forces the model to trade off data likelihood for constraint satisfaction, confirming the distributional bias observed in Table 1. $\qquad\square$

## B DARCY FLOW

*Darcy Flow* refers to a set of equations used to study the (laminar) flow of fluids in porous media. In two dimensions it is defined by the following set of equations for $\boldsymbol{x} = (x, y)$ Schlichting & Gersten (2017); Jacobsen et al. (2025); Bastek et al. (2025):

$$\boldsymbol{u}(\boldsymbol{x}) = - K(\boldsymbol{x})\nabla p(\boldsymbol{x}) \tag{20}$$

$$\nabla \cdot \boldsymbol{u} = f(\boldsymbol{x}) \tag{21}$$

$$\boldsymbol{u} \cdot \hat{\boldsymbol{n}} = 0, \text{ boundary condition} \tag{22}$$

$$\int p(\boldsymbol{x})\,\mathrm{d}\boldsymbol{x} = 0, \tag{23}$$

where $K$ is a permeability field that describes how easy a fluid flows through the medium, $p$ is the pressure field that defines where the fluid is pushed and pulled, $\boldsymbol{u}$ the velocity field of the fluid (as visualized in Fig. 4) and $f$ is the net flow of fluid through a given point. *Net* flow means that if there is the same amount fluid entering and exiting at a given point, then the net flow is zero. As a more concrete example, we can use Darcy flow to describe how water will flow through a body of sand. We can expect more water to flow at the areas where we apply more pressure to squeeze out the water.

There are many important assumptions around Darcy flow before it is applicable. We must have laminar flow at steady state, which means that there must not be sudden changes in the flow of the fluid. This depends on the geometry of the body that the fluid is traversing through, its viscosity, its speed, among other factors that are generally described through the Reynolds number Schlichting & Gersten (2017); Potter & Ramadan (2012). The Reynolds number needed to break into critical flow also will change between different configurations, so it may be difficult to know a priori if Darcy flow is an appropriate way to model the system.

Through experience it has been observed that the Darcy flow approximation can work well for media that are "fine-grained" enough, as the gaps between the particles in the porous matrix do not break the flow of water Woessner & Poeter (2020). However, at a big enough particle size, water starts to bounce more between collisions, causing the flow to become more turbulent.

### B.1 QUALITATIVE RESULTS FROM DARCY FLOW

Fig. 5 shows a histogram of the learned distribution of values for pressure and permeability using each of the compared methods. Particularly, as noted by (Bastek et al., 2025), PIDM presents excessive bias compared to the other methods.

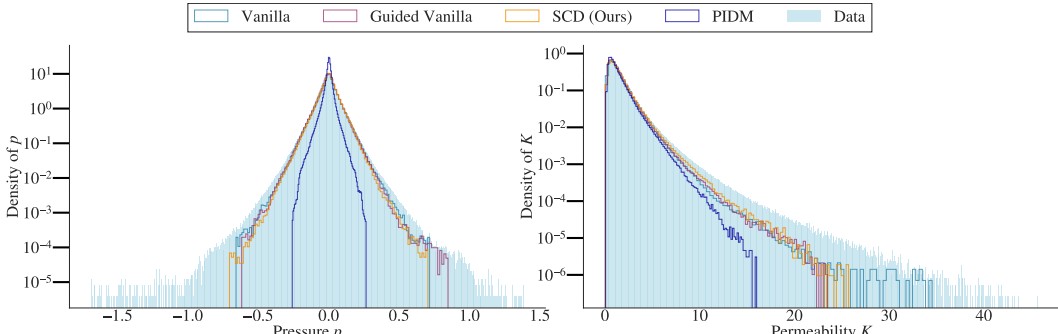

Figure 5: Darcy flow validation. 1000 samples were generated with each approach, using 100 denoising steps for each. Error bars indicate the minimum and maximum from the samples. The distribution PIDM learns is highly biased; all other methods have higher distributional fidelity. Particularly, our method shows low distributional bias compared to vanilla.

Qualitative samples using each of the methods can be seen in Fig. 6.

## C  LOSS-GUIDED DIFFUSION MODELS

*Loss-Guided Diffusion Models* were proposed by Song et al. (2023b) as a "plug-and-play" guidance method to make pre-trained diffusion models generate better constraint-compliant samples at inference time. The core idea is that a learned unconditional score $s_\theta^* \approx \nabla_{\boldsymbol{x}_t} \log p(\boldsymbol{x}_t)$ acts as a prior to which we can apply Bayes' rule for a posterior $p(\boldsymbol{x}_t|c)$ with some condition $c$:

$$\nabla_{\boldsymbol{x}_t} \log p\left(\boldsymbol{x}_t | c\right) = \nabla_{\boldsymbol{x}_t} \log \left( \frac{p\left(c | \boldsymbol{x}_t\right) p\left(\boldsymbol{x}_t\right)}{p(c)} \right) \tag{24}$$

$$= \nabla_{\boldsymbol{x}_t} \log p\left(c | \boldsymbol{x}_t\right) + \nabla_{\boldsymbol{x}_t} \log p\left(\boldsymbol{x}_t\right) - \underbrace{\nabla_{\boldsymbol{x}_t} \log p(c)}^{0} \tag{25}$$

$$= \nabla_{\boldsymbol{x}_t} \log p\left(c | \boldsymbol{x}_t\right) + \nabla_{\boldsymbol{x}_t} \log p\left(\boldsymbol{x}_t\right), \tag{26}$$

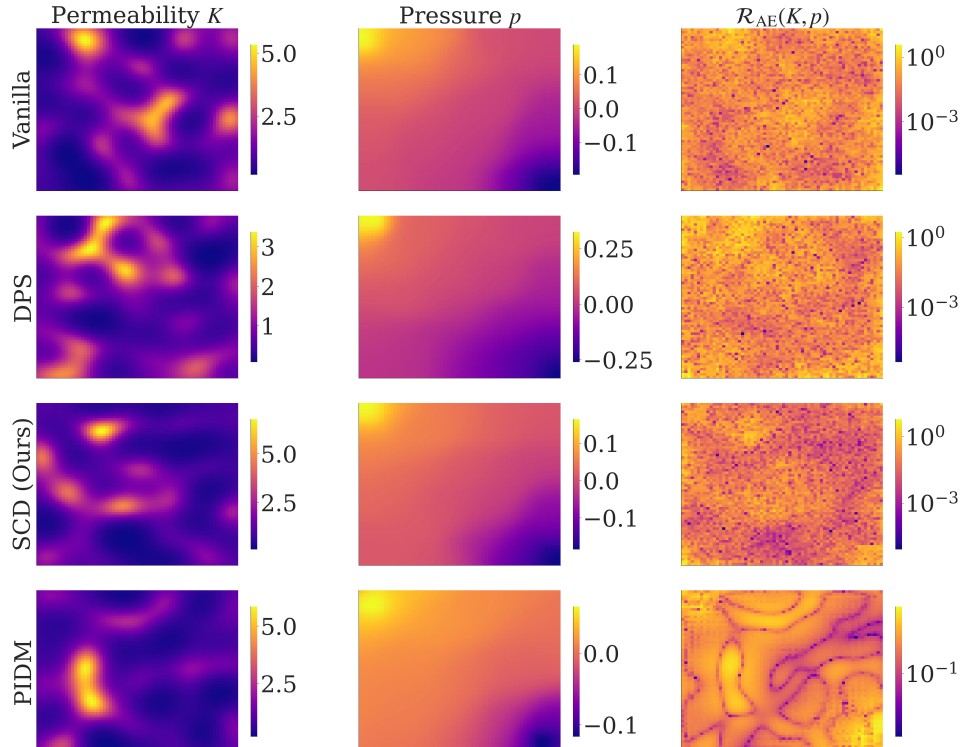

Figure 6: Residuals and samples produced by vanilla diffusion, DPS guidance, SCD and PIDM.

where $\boldsymbol{x}_t$ is the diffusion sample at time $t$ and $c$ is a condition for the generative process, and we have used the fact that $\log p(c)$ is a constant so its gradient is 0. The likelihood term $p(c|\boldsymbol{x}_t)$ is analytically untractable, as the condition $c$ only applies to clean samples $\boldsymbol{x}_0$. However, we can write it as follows:

$$p(c|\boldsymbol{x}_t) = \int_{\boldsymbol{x}_0} p(c|\boldsymbol{x}_0, \boldsymbol{x}_t) p(\boldsymbol{x}_0|\boldsymbol{x}_t) \mathrm{d}\boldsymbol{x}_0 = \int_{\boldsymbol{x}_0} p(c|\boldsymbol{x}_0) p(\boldsymbol{x}_0|\boldsymbol{x}_t) \mathrm{d}\boldsymbol{x}_0, \tag{27}$$

where we have assumed that $c$ is conditionally independent of $\boldsymbol{x}_t$ when given $\boldsymbol{x}_0$. We then take the approximation:

$$p(\boldsymbol{x}_0|\boldsymbol{x}_t) \approx \mathcal{N}\left(\boldsymbol{x}_0|\mu, \Sigma\right), \tag{28}$$

with mean parameter $\mu$ and covariance parameter $\Sigma$, as this allows us to use Tweedie's formula, connecting the score function with the exact moments of $p(\boldsymbol{x}_0|\boldsymbol{x}_t)$ (Efron, 2011; Rissanen et al., 2025):

$$\mu = \mathbb{E}[\boldsymbol{x}_0|\boldsymbol{x}_t] = \boldsymbol{x}_t + \sigma_t^2 \nabla_{\boldsymbol{x}_t} \log p(\boldsymbol{x}_t) \tag{29}$$

$$\Sigma = \mathbb{C}\text{ov}[\boldsymbol{x}_0|\boldsymbol{x}_t] = \sigma_t^2 \left( \sigma_t^2 \underbrace{\nabla_{\boldsymbol{x}_t}^2 \log p(\boldsymbol{x}_t)}_{\text{Hessian}} + \mathbf{I} \right). \tag{30}$$

Since the Hessian is expensive to evaluate in high-dimensional data, it is common to see the approximation $\Sigma \approx \sigma_{0|t}^2 \mathbf{I}$ be used, and this is the choice taken by Song et al. (2023b). Then, we can assume that the likelihood on clean samples $p(c|\boldsymbol{x}_0)$ can be expressed with a differentiable, lower-bounded constraint loss function $\ell_c : \mathcal{X}_0 \to \mathbb{R}$ by:

$$p(c|\boldsymbol{x}_0) = \frac{\exp\left(-\ell_c(\boldsymbol{x}_0)\right)}{Z}, \tag{31}$$

where $Z = \int_{\boldsymbol{x}_0} p(\boldsymbol{x}_0) \exp\left(-\ell_c(\boldsymbol{x}_0)\right) \mathrm{d}\boldsymbol{x}_0$ is a normalizing constant. Plugging equation 31 into equation 24, we get:

$$\nabla_{\boldsymbol{x}_t} \log p\left(\boldsymbol{x}_t|c\right) = \nabla_{\boldsymbol{x}_t} \log p\left(\boldsymbol{x}_t\right) + \nabla_{\boldsymbol{x}_t} \log \int_{\boldsymbol{x}_0} \exp\left(-\ell_c(\boldsymbol{x}_0)\right) \mathcal{N}\left(\boldsymbol{x}_0; \mu, \sigma_{0|t}^2 \mathbf{I}\right) \mathrm{d}\boldsymbol{x}_0. \tag{32}$$

Song et al. (2023b) point out that Chung et al. (2023a)'s DPS is a special case of this equation taking $\sigma_{0|t} \to 0$ and with $\ell_c$ being a linear projection. Song et al. (2023b) then proposes the approximation to the integral by using Monte Carlo integration with $\sigma_{0|t}^2 = \frac{\sigma_t^2}{1+\sigma_{t^2}}$. We adopt this scheme with 16 Monte Carlo samples to guide the vanilla diffusion model for our Darcy Flow experiments as a baseline.

## D ARCHITECTURES AND TRAINING DETAILS

The architectures and experiments on this work were implemented using PyTorch 2.4.1. All experiments were ran on single GPUs, from either NVIDIA H200, H100, A100 or V100 GPUs. For both experiments, $\gamma_\theta$ is implemented as a 2-layer MLP with an embedding dimension of 100. The input for either task is the denoiser output $D_\theta(x_t, t)$ and the diffusion time $t$. Between each layer there is an ELU activation function to ensure that the scaling factor remains positive.

### D.1 MODIFIED LOSS FUNCTION

The loss function by Karras et al. (2022) has the form seen in Eq. (3) with the choice of distributions $t \sim \text{LogNormal}(\mu_{\text{train}}, \sigma_{\text{train}}^2)$ and $x_t \sim \mathcal{N}(x_0, t\mathbf{I})$, and weighting function $w(t) = (t + \sigma_{\text{data}^2})/(t\sigma_{\text{data}}^2)$ . Crucially, the choice for the distribution for $t$ is free, i.e. a design choice; Karras et al. (2022) choose the log-normal distribution based on the observation that most of the important denoising steps in a diffusion model happen in the "intermediate" noise levels, since at high noise levels there is little distinction between steps and at very-low noise levels the differences are negligible. However, on the ODE problems presented in this paper we have observed that low noise levels have a substantial effect on the final residuals. Simply trying to set $\mu_{\text{train}}$ to a small value can cause numerical stability issues, because it may start sampling values that are too small and consequently make the weighting function blow up.

Following this, and based on the observation that in practice most numerical integration samplers always end at a predetermined minimum time step, we choose the noise level distribution $t \sim \text{TruncLogNormal}(\mu_{\text{train}}, \sigma_{\text{train}}^2, a)$ where $a$ defines the lowest possible noise level to be sampled. With this change, we noticed a substantial improvement in the residuals in the Darcy Flow experiment, with the results shown in Table 4. We use a mean of -1.5 and standard deviation of 1.2 for the log-normal loss and a mean of -2, standard deviation of 1.7 and truncation lower limit of -4 for the truncated log-normal loss.

Table 4: Residuals obtained in the Darcy flow experiments using the noise level distribution $t \sim \text{LogNormal}(\mu_{\text{train}}, \sigma_{\text{train}}^2)$ by Karras et al. (2022) and our choice $t \sim \text{TruncLogNormal}(\mu_{\text{train}}, \sigma_{\text{train}}^2, a)$. Using the truncated log-normal shows a substantial improvement over the log-normal in this task.

| | | *Original* | *increasing misspecificication* $\longrightarrow$ | | |
|---|---|---|---|---|---|
| **Distribution** | **Method** | $f_{\max} = 10$ | $f_{\max} = 20$ | $f_{\max} = 30$ | $f_{\max} = 40$ |
| Log-normal | Vanilla | $0.874 \pm 0.411$ | $0.874 \pm 0.411$ | $0.874 \pm 0.411$ | $0.874 \pm 0.411$ |
| | Guided Vanilla | $0.869 \pm 0.382$ | $0.862 \pm 0.372$ | $0.872 \pm 0.404$ | $0.869 \pm 0.379$ |
| | SCD (ours) | $0.342 \pm 0.136$ | $0.414 \pm 0.174$ | $0.429 \pm 0.173$ | $0.428 \pm 0.174$ |
| Truncated log-normal | Vanilla | $0.157 \pm 0.071$ | $0.157 \pm 0.071$ | $0.157 \pm 0.071$ | $0.157 \pm 0.071$ |
| | Guided Vanilla | $0.141 \pm 0.063$ | $0.139 \pm 0.067$ | $0.140 \pm 0.065$ | $0.139 \pm 0.066$ |
| | SCD (ours) | $0.113 \pm 0.048$ | $0.106 \pm 0.049$ | $0.114 \pm 0.059$ | $0.118 \pm 0.051$ |

Following these results, we use the truncated log-normal distribution for all our experiments.

### D.2 CIRCLES

For the toy example we use a 3 layer MLP with an embedding dimension of 128. To train the networks we use the Adam optimizer with $\beta_1 = 0.9$, $\beta_2 = 0.999$ and a fixed learning rate of $\alpha = 1 \cdot 10^{-4}$.

The data set consisted of 10000 points sampled from the unit circle, and the models were trained over 1000 epochs with a batch size of 128.

For the "Dent" variant of misspecification, we use the following parametric curve:

$$C(\theta) = (r(\theta)\cos(\theta), r(\theta)\sin(\theta)) \tag{33}$$

$$r(\theta) = 1 - 0.25 \cdot \beta\left(\frac{\mathrm{wrap}(\theta - \frac{\pi}{2})}{1.2}, 5\right) \cdot \left(1 + 0.6\left(1 - 2\left(\frac{\mathrm{wrap}(\theta - \frac{\pi}{2})}{1.2}\right)^2\right)\right) \tag{34}$$

$$\mathrm{wrap}(\theta) = ((\theta + \pi) \mod 2\pi) - \pi \tag{35}$$

$$\beta(u, 5) = \begin{cases} (1 - u^2)^5 : |u| < 1 \\ 0 \text{ otherwise}, \end{cases} \tag{36}$$

where C defines the coordinates of every point in the curve in polar coordinatese.

### D.3 DARCY FLOW

For Darcy Flow we used the UNet implementation by Karras et al. (2023). We use the Heun sampler implementation by Karras et al. (2022) with 100 denoising steps. The used hyperparameters on the network are summarized on Table 5.

Table 5: Architecture hyperparameters for the Darcy Flow experiments

| Hyperparameter | Value |
|---|---|
| Model channels | 24 |
| Number of residual blocks | 8 |
| Per-resolution multipliers | [1, 2, 3, 4] |
| Attention resolutions | [16, 8] |

### D.4 HELMHOLTZ EQUATION

For the Helmholtz Equation we used the UNet implementation by Karras et al. (2023). We use the Heun sampler implementation by Karras et al. (2022) with 100 denoising steps. The used hyperparameters on the network are summarized on Table 6.

Table 6: Architecture hyperparameters for the Helmholtz Equation experiments

| Hyperparameter | Value |
|---|---|
| Model channels | 32 |
| Number of residual blocks | 8 |
| Per-resolution multipliers | [1, 2, 3, 4] |
| Attention resolutions | [16, 8] |

### D.5 RUNTIMES

The details on the runtimes on an NVIDIA H200 GPU with vanilla diffusion and our method are shown in Table 7. We note that our method sees the most impact at sampling time, since the overhead duplicates per each sampling iteration because the Heun sampler makes two neural function evaluations per iteration.

The baseline PIDM results were reproduced with a pre-trained model provided by Bastek et al. (2025), and for the misspecification experiments we trained a diffusion model with PIDM with each misspecified residual using the same hyperparameter setups as Bastek et al. (2025). The vanilla score-matching and SCD networks were trained using Adam with an initial learning rate of $2 \cdot 10^{-3}$ and weight decay as detailed by Karras et al. (2023). Both the vanilla model and SCD were trained for 300k iterations, although SCD plateaus at around 250k. We use a batch size of 64. For all experiments we use Exponentially Moving Averages for sample generation with a decay rate of 0.99.

Table 7: Runtimes of vanilla and our method on training and sampling on an NVIDIA H200 GPU. Sampling is done for eight samples at a time using the Heun sampler implementation by Karras et al. (2022). Numbers are reported as a mean with two standard errors over 300 iterations. Units of time are specified for each column.

| Method | Training iteration wall clock time (ms) | Sampling iteration wall clock time (s) |
|---|---|---|
| Vanilla | $110 \pm 3.62$ | $7.79 \pm 0.021$ |
| SCD (ours) | $148 \pm 2.25$ | $10.3 \pm 0.029$ |

## E    ADDITIONAL "CHOP" SAMPLES

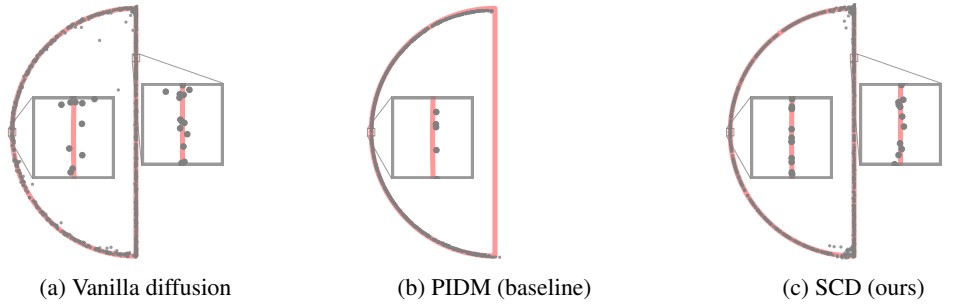

(a) Vanilla diffusion                    (b) PIDM (baseline)                    (c) SCD (ours)

Figure 7: Additional samples from the "chop" example.

## F    DISCLOSURE OF THE USE OF LARGE LANGUAGE MODELS

In this paper, LLMs were used only for minor grammatical edits, word polishing, or rephrasing. They did not contribute to research ideation, experiments, or core writing. All suggestions from LLMs were manually verified and edited by the authors prior to final inclusion. Additionally, LLMs were used as coding assistants as part of implementing the methods in code.

