# OpenReview forum: "Softly Constrained Denoisers for Diffusion Models"
_ICLR.cc/2026/Conference — Submitted to ICLR 2026_

### Official Review · Reviewer_V5bD · 2025-10-19

**Soundness:** 3
**Presentation:** 3
**Contribution:** 2
**Rating:** 4
**Confidence:** 3

**Summary:**

The paper proposes Softly Constrained Denoisers (SCD) to embed constraint knowledge directly into diffusion model denoisers via a learnable gradient-based adjustment, instead of adding explicit regularizers or inference-time guidance. This gives diffusion models a soft inductive bias toward constraint satisfaction while maintaining data fidelity and robustness to misspecified constraints. Results on toy and PDE (Darcy flow) tasks show SCD improves constraint compliance without the strong bias seen in Physics-Informed Diffusion Models (PIDM).

**Strengths:**

1. The idea of integrating constraints through architecture, not loss is interesting.

2. The demonstrated robustness can handle constraint misspecification adaptively.

3. Minimal computational cost, drop-in compatible with standard denoisers.

**Weaknesses:**

1. Although the proposed method is more flexible than traditional physics-informed regularization, it still relies on the correctness of the constraint function (for example, the residual term that encodes a physical law). If that constraint is inaccurate or incomplete, the model can still be guided in the wrong direction. The paper argues that the learnable scaling factor can help the model “ignore” bad constraints, but there is no formal guarantee that this will always happen. In highly misspecified cases, the denoiser may still learn unrealistic or unstable patterns. More importantly, PIDM can also be coupled with an adaptive weight factor to alleviate bad constraints. I do not see the inherent differences if the proposed method applies to real-world scenario.

2. PIDM or PINN usually applies to scenario genuinely governed by PDE or physical laws. For PINNs or PIDMs, the constraint corresponds to a real physical model, so enforcing it makes scientific sense, even if imperfect. For SCD, the “constraint” could be arbitrary or only partially relevant to the data; so, when the data is not truly governed by a known PDE, the method’s rationale becomes weaker.

3. The core formulation of the softly constrained denoiser is derived from several simplifying assumptions that are not theoretically rigorous. The authors replace complex probabilistic terms with direct gradient-based corrections and diagonal approximations. While this makes the approach practical, it weakens the theoretical grounding. As a result, it is unclear whether the denoiser still accurately represents the true underlying diffusion process or only an approximation that happens to work empirically.

4. The experiments are restricted to simple two-dimensional toy problems and the Darcy flow benchmark. These are clean and controlled settings with relatively smooth constraints. The paper does not test high-dimensional, noisy, or real-world datasets. It also lacks ablations showing how different definitions of the constraint function or the scaling factor affect performance. This leaves open questions about robustness and generalization beyond the presented cases. Moreover, If SCD is designed for scenarios not entirely governed by physical laws, the experiments should reflect that. For example, weather forecasting is not entirely governed by diffusion and advection and usually needs post-hoc assimilation. There are currently no convincing experiments to reflect scenarios where physical law alone is not perfect.

**Questions:**

To strengthen the paper, the authors may want to address the following questions:

1. How does SCD behave with non-differentiable or discontinuous constraints?

2. Can $\gamma_\theta$ be learned dynamically to detect and suppress harmful constraints?

3. How stable is training when $r_c(x)$ provides noisy or conflicting gradients?

4. Could this framework extend to hard constraints (e.g., manifold diffusion)?

5. How does SCD scale with high-dimensional scientific data (e.g., 3D turbulence)?

---

> ### Author Response · Authors · 2025-11-22
> **Answer to reviewer V5bD**
>
> We thank the reviewer for their thoughtful review and their time in the review process. We answer the comments below:
>
> “*...The authors replace complex probabilistic terms with direct gradient-based corrections and diagonal approximations… As a result, it is unclear whether the denoiser still accurately represents the true underlying diffusion process or only an approximation that happens to work empirically.*”
>
> We believe there may have been a misunderstanding of how the method works: Even though we motivate the denoiser design using an approximation of the exact guidance formula, the resulting method is *not* an approximation that happens to work empirically. It has the exact same asymptotic guarantees as standard diffusion models have: In the limit of infinite data and reaching the global optimum of the training loss, the model will converge to the data distribution. This *is* important, as convergence to the data distribution is a standard property of almost all generative frameworks, and allows us to reason about the behaviour of the model.
>
> The reason that the denoiser represents the true underlying data distribution is that it is trained end-to-end using the standard denoising score matching loss. We have now included Algorithm 1 and a new Figure 2 to show the training process and the computational graph of the denoiser architecture and have added clarifying explanations in the Introduction and Methods sections. The key insight of the paper is that the denoiser architecture is the one design choice that the diffusion model paradigm does not constrain in any way, and we are free to embed additional inductive biases in it without breaking the asymptotic guarantees. A paramount example of this is the use of a UNet or a DiT as the backbone of the diffusion model.
>
> In contrast, PIDM does not have this guarantee due to changing the diffusion loss function, and we can correspondingly predict the failure mode in Figure 1, for example. To clarify how PIDM causes bias, we have added new Propositions 3.1. and 3.2. in lines 254 and 264 showing that 1) the optimal solution to the PIDM loss is not $E[x_0|x_t]$, dropping all data modelling guarantees 2) the diffusion ELBO for PIDM will be worse at optimality than for a diffusion model without a regularising term.
>
>
> “*...The paper argues that the learnable scaling factor can help the model “ignore” bad constraints, but there is no formal guarantee that this will always happen. In highly misspecified cases, the denoiser may still learn unrealistic or unstable patterns.*”
>
> To reiterate: In the sense that there is no formal guarantee that *anything in particular at all* happens in diffusion model training due to possibilities of local optima, we indeed do not have formal guarantees. In the sense of global optima, as outlined in the background section, we do have the exact same formal guarantees that a standard denoiser has, although practical generalisation is of course expected to change due to different inductive biases.
>
> “*More importantly, PIDM can also be coupled with an adaptive weight factor to alleviate bad constraints. I do not see the inherent differences if the proposed method applies to real-world scenario.*”
>
> We do not see how such a weight factor would help. If we, e.g., add a learnable positive constant to scale the regularization term in PIDM for each diffusion time level, a trivial solution to the optimization problem would be to zero out all of the terms, resulting in standard diffusion training. On the other hand, handling this constraint information *while optimizing for the DSM loss* incentivizes the model to actually use it as opposed to simply zero-ing it out to improve the optimization target. We are open to suggestions on how to do this with the PIDM framework, however.
>
> “*PIDM or PINN usually applies to scenario genuinely governed by PDE or physical laws. For PINNs or PIDMs, the constraint corresponds to a real physical model, so enforcing it makes scientific sense, even if imperfect.*”
>
> We agree that enforcing an imperfect constraint is *helpful*, but we argue that it is often not the optimal thing to do. In practice, physical descriptions of real complex systems are almost always approximations of the true phenomena, and we may also have imperfectly measured parameter values for the PDEs in practice. As such, we argue that it is useful to have the capability to smoothly use the constraint information when useful and ignore it when the data disagrees. A particular case where a similar idea has worked well is Finzi et al.'s Residual Pathway Priors [1], where a *soft inductive bias* helps dealing with slight deviations from geometric constraints. Regarding PINNs, we argue that they are not directly comparable to the data-based generative setup of PIDM and SCD, and instead focus on competing with traditional PDE solvers.

---

> > ### Author Response · Authors · 2025-11-22
> > **Continuing answer to reviewer V5bD**
> >
> > “*For SCD, the “constraint” could be arbitrary or only partially relevant to the data; so, when the data is not truly governed by a known PDE, the method’s rationale becomes weaker.*”
> >
> > We hope that the previous answers addressed this issue, but we are open to discuss more.
> >
> > **Questions**:
> >
> > “*How does SCD behave with non-differentiable or discontinuous constraints?*”
> >
> > We have added a description on the Methods section regarding how we treat the constraint functions in general, where we generally assume access to some differentiable function $\mathcal{R}$ that encodes it. In practice, in frameworks such as PyTorch it is also possible to handle discontinuities through the use of sub- or super-gradients depending on the convexity or concavity around the point that is evaluated (details in [the PyTorch documentation](https://docs.pytorch.org/docs/stable/notes/autograd.html#gradients-for-non-differentiable-functions)), and this is exactly what happens with functions like the absolute value and the ReLU around their discontinuity points. Thus, if there are a finite number of discontinuities, it would still be possible to build useful relaxed constraint functions to differentiate through with our framework.
> >
> > "*Can theta_gamma be learned dynamically to detect and suppress harmful constraints?*"
> >
> > This is part of our motivation for integrating such a learnable scaling factor, and we believe the example shown in Figure 1 is in fact an example where we see this in action.
> >
> > "*How stable is training when r_c provides noisy or conflicting gradients?*"
> >
> > We believe our misspecification experiments in Darcy Flow shows an example of conflicting gradients, where the gradient information carries information counter to what is observed in the data. We have also injected noise in the gradient for the Helmholtz Equation experiments, and we witnessed no significant change in the training dynamics in neither case.
> >
> > "*Could this framework extend to hard constraints (e.g., manifold diffusion)?*"
> >
> > Our method does not impose a particular structure on the network used or on the sampling process. In the work we have shown experiments using both UNets and MLPs, and in a similar vein if we wished so we could use other equivariant variants of neural network architecture to enforce such constraints in the outpuut. In examples such as manifold diffusion, the constraints are handled on the sampling process itself with no need to address the neural network architecture, as we mention in the Related Work section.
> >
> > "*How does SCD scale with high-dimensional scientific data (e.g., 3D turbulence)?*"
> >
> > We argue that we *are* in a relatively high-dimensional setting as we deal with samples in $\mathcal{R}^{n\times n}$, with $n=64$ in Darcy Flow and $n=128$ in the newly added experiments for the Helmholtz Equation. As we observed in the time measurements for Darcy Flow, the additional overhead using our method is relatively small for each denoising step.
> >
> > We hope that the answers have been satisfactory, and hope to hear if the reviewer has additional comments or questions.
> >
> > ## References
> >
> > [1] Finzi, M., Benton, G. and Wilson, A.G., 2021. Residual pathway priors for soft equivariance constraints. Advances in Neural Information Processing Systems, 34, pp.30037-30049.

---

> ### Comment · Reviewer_V5bD · 2025-11-22
>
> Thanks for the long rebuttal, but I feel like a bit talking past each other.
>
> For the first point I have raised, for example, I am still not entirely sure where and how a formal guarantee is secured or demonstrated in this paper. If I remember correctly, you were using a scaling on the constraints so the learning process can be more or less affected by the constraints as you like, right? I am not really concerned that your method would lose the global optimal or anything. My worry is simply that this modification is too trivial and not novel enough. I saw all other reviewers feel the same way, so I am a bit surer I am not being too mean to authors on what is novel versus not novel.
>
> One of your points is quite interest. "In practice, physical descriptions of real complex systems are almost always approximations of the true phenomena, and we may also have imperfectly measured parameter values for the PDEs in practice." I both agree and disagree. The point I am disagree with is that, I believe, if the problem is so complex that existing PDEs cannot describe sufficiently, then we should not use PIDM or PINN at all! I really do not believe this can be solved by just losing the physical constraints and let model learn more in the data-driven manner.
>
> For example, in climate modeling, the governing equations are inherently imperfect, which is why data assimilation has always been essential. Deep learning approaches now attempt to model climate dynamics in a fully data-driven manner—and that is acceptable. Some recent works combine deep learning with physical principles, but they do so carefully: physics is not simply an afterthought or a weak regularizer. Instead, these models are designed so that the learned dynamics respect fundamental physical constraints without imposing rigid PDE formulations. So, to put it simple, the reason I recommended rejection to this paper is in the inherent simplicity and lack of novelty of the proposed solution. Balancing loss function by a learnable scaler, which is at least what appears to be the core contribution, seems not quite so impressive enough.

---

> > ### Author Response · Authors · 2025-11-24
> >
> > We thank the reviewer for the engagement and honest interest in the discussion. Novelty was **not** pointed out as a weakness in the initial review, but rather our approach "integrating constraints through architecture, not loss" **was pointed out as a strength**. However, there still seems to be a misunderstanding about the core method, which is what we tried to clarify from different angles with the long rebuttal. We believe that this is especially important since the reviewer now states that the main reason for a negative score is novelty. To pin down what we see the reviewer states the method as being is:
> >
> >  *"Balancing loss function by a learnable scaler, which is at least what appears to be the core contribution"*
> >
> > We are explicitly **not** balancing the loss function or changing it in any way. This is indeed the core insight of the method: Such changes would cause bias that is theoretically easy to show. Post-hoc guidance methods also cause bias. In contrast to them, our method does not cause such theoretically clear-cut bias relative to a standard diffusion model. There is no need for a specific proof of this, since this is a property of standard diffusion models and there is no reason why we would not have it as well. We do show that PIDM does not have this property, however, giving us a theoretical advantage over PIDM.
> >
> > We are happy to discuss the novelty and simplicity of the method further, but this requires us to agree and explicitly remember what the claimed core contribution is. Otherwise, it seems to us contradictory to say that the method is trivial or too simple. In short:
> > - **We change the architecture to include constraint information in a specific manner inspired by the guidance literature. This is similar to the adaptive "Residual Pathway" we cite by Finzi et al.**
> >
> > This was not considered in PIDM or in the work using guidance methods directly for enforcing constraints, and we claim that is more theoretically grounded than PIDM or guidance. We include new visualisations and an algorithm in the revised paper.
> >
> > **On the points about using physical constraints vs. learning from data.**
> >
> > *"... then we should not use PIDM or PINN at all! I really do not believe this can be solved by just losing the physical constraints and let model learn more in the data-driven manner."*
> >
> > We do not lose the physical constraints: We use them to improve constraint satisfaction. The result is not a hard constraint, but this equally true of PIDM. Regarding climate: Our work is, in fact, a particular way to introduce physical constraints into the model! We focus deliberately on developing a new methodological component for diffusion models instead of weather applications in particular. An application to weather would be exciting future work.
> >
> > *"Some recent works combine deep learning with physical principles, but they do so carefully: physics is not simply an afterthought or a weak regularizer."*
> >
> > Is the claim that in domains such as climate, using a method such as ours would be less careful than some competing methods? It seems like it would help if these specific methods and comparisons to them, especially highlighting how our method could be considered less careful, were pointed out.
> >
> > Thank you again for the discussion, and we appreciate hearing about additional comments or concerns.

---

> > > ### Author Response · Authors · 2025-11-27
> > >
> > > We thank the reviewer for reassessing their score from 4 to 6, and would be happy to hear if there are any additional questions.

---

### Official Review · Reviewer_N3A7 · 2025-10-30

**Soundness:** 1
**Presentation:** 2
**Contribution:** 1
**Rating:** 2
**Confidence:** 3

**Summary:**

Softly Constrained Denoiser (SCD) proposes to embed a constraint directly into the denoiser as a guidance-style correction. Advantage against PIDM is claimed in terms of misspecification robustness and distributional bias.

**Strengths:**

On toy example of data generation on a circle with misspecified constraints, SCD maintains low Wasserstein-1 distance (i.e., better data fidelity) while PIDM degrades as misspecification increases, indicating robustness of SCD to wrong constraints. On the Darcy Flow PDE benchmark, SCD avoids the strong distributional bias observed with PIDM.

**Weaknesses:**

The theoretical idea is not sufficiently novel. The experiments are limited to a toy setup of circles and a single PDE benchmark, so external validity across other constraint types (hard equalities/inequalities, manifold constraints) is unclear, and distributional fidelity is mostly argued via histograms/qualitative plots rather than likelihood/FID score. Methodologically, the denoiser correction hinges on aggressive approximations which avoids VJPs but offers no guarantee against bias or consistency. The paper does not contain extensive theoretical analysis or sufficient experimental results to clear the high bar of ICLR.

**Questions:**

None

---

> ### Author Response · Authors · 2025-11-22
> **Answer to reviewer N3A7**
>
> We thank the reviewer for their time in the review process. We respond to the comments one-by-one below.
>
> “*Methodologically, the denoiser correction hinges on aggressive approximations which avoids VJPs but offers no guarantee against bias or consistency.*”
>
> We believe that there may have been a misunderstanding of how the method works. We *do explicitly* offer a guarantee against bias and have a proof of consistency in exactly the same way that a standard diffusion model without explicit constraint information at all has them, as explained in the background section. The reason that our method does not have any more bias than standard diffusion models is that we *are not* proposing an approximate inference-time modification to the sampling process, and we are not changing the diffusion loss function. We instead propose a method for deriving architectures that embed inductive biases provided by a constraint function. Instead of proposing such architectures arbitrarily, we base the embedded inductive bias on a principled approximation of the guidance formula, highlighted in Equations 6-9. We could go with other options as well, but we find that our choice strikes a good balance between embedding a sufficient inductive bias and computational efficiency. We realise now that the core idea of the method was not stressed sufficiently in the paper, and have updated text in the introduction and methods sections to highlight it further. We have also included Algorithm 1 and a new Figure 2 to showcase the training process and the denoiser architecture computational graph.
>
> “*external validity across other constraint types (hard equalities/inequalities, manifold constraints) is unclear*”
>
> The residuals we use *are* equality constraints, and to integrate inequality constraints we can follow the method mentioned by Bastek et al. where the constraint is evaluated in a ReLU, effectively turning the inequality constraint into an equality constraint. The example of the 2D circle is a manifold constraint as well, as the data lies on a subset of the full 2-dimensional space. Regarding *hard constraints*: Our proposed method does not impose a specific structure to be used in the base denoiser. Thus, if we wish to handle, e.g., equivariance constraints, we can use an equivariant architecture as the backbone of our denoiser, or follow the methods we mention in the **Related Work** in the *Hard-Constraint Diffusion via Modified Dynamics* section, e.g. [1,2,3], where the *diffusion sampling process* is modified to enforce geometric constraints with no need to address the neural architecture itself, where the constraints are imposed through the denoising process and not through the network architecture.
>
> To address the experimental section weakness, we have included additional experiments on the Helmholtz Differential Equation, including an ablation on observation noise on the wave number, following FunDPS. We have also included DPS-based guidance baselines on the experiments. To support the bias measurements, we include NLL measurements on the test set for the compared methods. We have additionally added two propositions about the bias caused by regularisation-based methods on diffusion models in the main text. As our method optimizes only against the Denoising Score Matching loss function, which is shown to be unbiased and consistent in Song et al. [4], we maintain the same asymptotic convergence guarantees.
>
> We again thank the reviewer for their time and hope to hear if there are additional comments or questions.
>
> ## References
>
> [1] Valentin De Bortoli, Emile Mathieu, Michael Hutchinson, James Thornton, Yee Whye Teh, and Arnaud Doucet. Riemannian score-based generative modelling. In Advances in Neural Information Processing Systems 35 (NeurIPS), pp. 2406–2422, 2022.
>
> [2] Chin-Wei Huang, Milad Aghajohari, Joey Bose, Prakash Panangaden, and Aaron C Courville. Riemannian diffusion models. In Advances in Neural Information Processing Systems 35 (NeurIPS), pp. 2750–2761, 2022.
>
> [3] Ricky TQ Chen and Yaron Lipman. Flow matching on general geometries. In International Conference on Learning Representations (ICLR), 2024.
>
> [4] Yang Song, Jascha Sohl-Dickstein, Diederik P. Kingma, Abhishek Kumar, Stefano Ermon, and Ben Poole. Score-Based Generative Modeling through Stochastic Differential Equations. In International Conference on Learning Representations (ICLR), 2021.

---

> > ### Author Response · Authors · 2025-11-27
> >
> > We again thank the reviewer for their time and comments. As detailed in the point-by-point response above, we have revised the manuscript to address their concerns regarding novelty and have clarified what our core contribution to the diffusion model literature is: that is, a novel denoiser architecture that integrates physical constraints as a *soft inductive bias*, and not only a modification for the diffusion sampling process. Additionally, we have expanded the evaluations with NLL measurements to support our claims about bias, and have added an additional experiment on the Helmholtz equation to show our method is still robust with noisy gradients, as well as to illustrate the issues that come with imbalanced guidance. We believe these revisions have significantly improved the manuscript. Since the discussion period is ending, we respectfully request that the reviewer reassess the score based on these changes, and would be happy to hear if there are any remaining questions about our work.

---

### Official Review · Reviewer_iiNj · 2025-11-02

**Soundness:** 3
**Presentation:** 3
**Contribution:** 2
**Rating:** 2
**Confidence:** 4

**Summary:**

The paper addresses the problem of constraint misspecification in diffusion-based generative models. During inference, guidance typically approximates $P(x_0|x_t)$, leading to cumulative errors that significantly distort the final output distribution. On the other hand, applying direct regularization to the training loss tends to break the intrinsic connection between the denoiser and the score function, further biasing the modeled distribution. To mitigate these issues, the authors propose a softly constrained denoiser, which replaces the covariance term with a lightweight learnable scaling factor network. This design eliminates the need for expensive vector–Jacobian computations during guidance, while maintaining compatibility with standard diffusion training and sampling pipelines.

**Strengths:**

The paper offers a new perspective on constrained distribution estimation by introducing a learnable scaling factor to replace the covariance term. This substitution effectively reduces computational complexity while preserving the structure of standard diffusion training and sampling schemes, requiring no architectural or procedural modifications.

**Weaknesses:**

1. The method builds heavily on prior work, particularly on the approximation from Eq. (6) to Eq. (4), and the main contribution lies in substituting the covariance with a learnable scaling factor. While this reduces computation, it also diminishes interpretability and controllability. There is no analysis about the relationship between the scaling factor and covariance.

2. The experimental section is weak. The experiments in Fig 1 and Fig 2 are performed only on toy datasets. The quantitative results in Table 2 are not compelling; the improvements are marginal and fail to convincingly demonstrate the method’s practical utility. The claimed advantages of the softly constrained denoiser in the PDE setting are not well supported by the flow-based experimental results.

**Questions:**

1. The design of $r_c(x)$ and the residual function appears to be nontrivial. Could the authors provide a more systematic or generalizable recipe for constructing these functions?

2. Why were there no comparisons against guided vanilla diffusion or other standard benchmark methods in the relative works to more clearly establish the benefits of the proposed approach?

3. In Fig 1, 2 & 5, why don't show the results of guided diffusion?

---

> ### Author Response · Authors · 2025-11-22
> **Answer to reviewer iiNj**
>
> We thank the reviewer for the feedback. We would like to start by emphasizing that what we claim to be our main contribution is not a learnable scale or learnable covariance term for guidance methods, but *the integration of a guidance-like term into the denoiser architecture as a soft inductive bias*. The difference is that we train end-to-end through the entire computational graph, and as such gradient descent is free to push the architecture to use the guidance term simply whenever it is useful, and it is not forced to approximate a particular covariance matrix. The intent of the guidance derivations was to show a principled way to derive such softly biased architectures: The hypothesis was that since the addition of the exact guidance term at inference time would make any base denoiser comply with the constraint, directly introducing even an approximate version of the guidance term into the architecture directly would help finding denoiser parameters that generate compliant samples. We now realize that this was not explained explicitly enough in the paper, and have included a new Algorithm 1 and Figure 2 to show exactly how the training process and denoiser forward pass works. We have also changed the explanations in the introduction and methods section to highlight this further.
>
> In the following, we go through the rest of the comments in order.
>
> **Weakness 1**:
>
> “*While this reduces computation, it also diminishes interpretability and controllability*”
>
> We agree that inference-time guidance methods do have advantages, mainly the ability to switch the constraint function without retraining. This paper, however, focuses on the setting where we have predefined (approximate) constraints, and the data is within the bounds of these constraints. In this case, there is no need to switch the constraint at inference time, and instead the goal is to extract as much information as possible from the analytical constraint to improve sample and distributional quality. Further, we could combine our denoiser with any inference-time guidance method for additional post-hoc constraints.
>
> “*There is no analysis about the relationship between the scaling factor and covariance.*”
>
> Such analysis could be interesting in the sense of probing into the internals of the proposed denoiser, but we stress that *the scaling factor does not need to correspond with the covariance, and the scaling factor is not the only change to the learning process*. The precise way to read Eq.9 is that $D_{orig}$ learns to output values such that a linear combination of the original value and the gradient of $\log r_c$ at that value is trained to point towards $\mathbb E[x_0|x_t]$. As such, $D_{orig}$ is free to adjust as well.
>
> Drawing parallels with Finzi et al.’s Residual Pathway Priors [1], we treat this term as a free design component for a neural network architecture, not as a modification on the sampling process as opposed to guidance methods in general. To the best of our knowledge, this has not been explored before in the diffusion literature. We realize that this could have been explained better, so we have rewritten sections of the Introduction and methods on lines 76 and 174, please let us know if this clarifies what we mean.
>
> **Weakness 2**:
>
> To address the second weakness, we have included additional experiments on the Helmholtz Differential Equation, including an ablation on observation noise on the wave number,following FunDPS. We have also included DPS-based guidance baselines on the experiments. To support the bias measurements, we include NLL measurements on the test set for the compared methods.
>
> We also would like to emphasize that other related guidance-based methods have not focused on the fidelity-diversity tradeoff, instead focusing on reconstruction errors based on conditional generation at *inference time*, and to show this point we have included residual and NLL measurements on the Helmholtz Equation using FunDPS [2]. Our method is more directly comparable to *training time* methods in that we introduce the adjustment term as additional structure to the network while optimizing for the denoising score matching loss in a similar vein to Finzi et al.'s residual pathways between a geometrically constrained layer and unconstrained layer.
>
> Regarding the design of the $r_c$ function, for this work we have followed the recipe prescribed by Song et al. [3]: generally, if we have a function bounded from below, we are able to simply express $r_c$ as exp(-norm(loss(x))), and in our experiments we have found that the L1 norm works generally well, we have clarified this in the methods section, line 174. The L2 norm is more sensitive to outliers [2], which makes it unstable for the early denoising steps.

---

> > ### Author Response · Authors · 2025-11-22
> > **Continuing answer to reviewer iiNj**
> >
> > **Questions**:
> >
> > “*The design of r_c and the residual function appears to be nontrivial. Could the authors provide a more systematic or generalizable recipe for constructing these functions?*”
> >
> > We have added a clarification on how we do this in the methods section, line 174. The paper focuses on constraints that are possible to define as $\mathcal{R}(x)=0$, where $\mathcal{R}$ is some differentiable function. The residual is then formed simply as $||\mathcal{R}(x)||$, where $||\cdot||$ is some norm. We changed the notation to disentangle two uses of the original $r_c(x)$: The original binary constraint function $c(x)$, and the relaxed constraint function $l_c(x)$ used to calculate gradients from
> >
> > Please let us know if more clarification is needed.
> >
> > “*Why were there no comparisons against guided vanilla diffusion or other standard benchmark methods in the relative works to more clearly establish the benefits of the proposed approach?*”
> >
> > We have added comparisons against a DPS guidance implementation, showing that generally for these types of problems the improvements are generally not too significant, and if using imbalanced objectives they can cripple the distributional fidelity of the diffusion model.
> >
> > “*In Fig 1, 2 & 5, why don't show the results of guided diffusion?*”
> >
> > We have added these results to Figures 2 and 5 (now Figures 3 and 6). In Figure 1 we emphasize that our method is most comparable to other training-time methods, and would prefer to have the figures as big as possible for better visibility, though we are open to hearing points for adding the guidance results.
> >
> > We hope that our answers have brought some clarity to the questions, and hope to hear for more comments or questions from the reviewer.
> >
> > ## References
> >
> > [1] Finzi, M., Benton, G. and Wilson, A.G., 2021. Residual pathway priors for soft equivariance constraints. Advances in Neural Information Processing Systems, 34, pp.30037-30049.
> >
> > [2] Jiachen Yao, Abbas Mammadov, Julius Berner, Gavin Kerrigan, Jong Chul Ye, Kamyar Azizzadenesheli, and Anima Anandkumar. Guided diffusion sampling on function spaces with applications to PDEs. arXiv preprint arXiv:2505.17004, 2025.
> >
> > [3] Jiaming Song, Qinsheng Zhang, Hongxu Yin, Morteza Mardani, Ming-Yu Liu, Jan Kautz, Yongxin Chen, and Arash Vahdat. Loss-guided diffusion models for plug-and-play controllable generation. In Proceedings of the 40th International Conference on Machine Learning (ICML), volume 202 of Proceedings of Machine Learning Research, pp. 32483–32498. PMLR, 2023b

---

> > > ### Author Response · Authors · 2025-11-27
> > >
> > > We again thank the reviewer for their time and comments. As detailed in the point-by-point response above, we have revised the manuscript to address their concerns regarding novelty and have clarified what our core contribution to the diffusion model literature is: that is, a novel denoiser architecture that integrates physical constraints as a *soft inductive bias*, and not only a modification for the diffusion sampling process. Additionally, we have expanded the evaluations with NLL measurements to support our claims about bias, and have added an additional experiment on the Helmholtz equation to show our method is still robust with noisy gradients, as well as to illustrate the issues that come with imbalanced guidance. We believe these revisions have significantly improved the manuscript. Since the discussion period is ending, we respectfully request that the reviewer reassess the score based on these changes, and would be happy to hear if there are any remaining questions about our work.

---

### Author Response · Authors · 2025-12-01
**Closing Remarks**

In this paper, we propose Softly Constrained Denoisers (SCD), which embed constraint knowledge directly into the denoiser architectures for Diffusion Models, rather than using regularization terms in the loss or inference-time guidance. We systematically derive SCD inspired by the guidance literature, with the core contribution being that the Diffusion Model is able to exploit constraint knowledge without having to forfeit the theoretical guarantees from vanilla diffusion models, a key advantage over regularizer-based and guidance methods.

There have been some key issues throughout the rebuttal process. We summarize them and specify how we have addressed them point by point:

# Misunderstandings around the core contributions of the paper

One of the main concerns in this submission has been a misunderstanding of our key contributions. To reiterate, **our key contribution is the introduction of a principled way to introduce constraint knowledge to a Diffusion Model through the denoiser's architecture**. However, it seems like most reviewers believed that our claimed contribution is the introduction of a learnable, adaptive weight on the guidance term *used only at inference time*. We clarify that, although we have not seen previous literature use such an adaptive weight, **this is not our core contribution**, but rather an additional "knob" we introduce to let the architecture learn flexibly from the constraint information. To address this concern, we have:

- Clarified what we claim our main contributions to be in the Introduction, at line 76.
- Added Algorithm 1 and a new Figure 2 in the Methods section to clearly and visually explain our approach.
- Clarify that we **do not** drop the constraint information, a brief concern that reviewer V5bD shared in their reply to our rebuttal. Much to the contrary, we embed it directly into the denoiser architecture.

# Concerns about bias claims with our method

There were persisting concerns about the bias claims with our presented method, mainly stemming from the misunderstanding of our presented approach. To support our claims, we have:

- Added propositions 3.1 and 3.2 in Section 3.1, at line 239 and with detailed proofs in the Appendices, to show that regularizer-based methods are theoretically biased, thus forfeiting the convergence guarantees to the optimal denoiser.
- Added NLL measurements in the experiments to support our bias claims against the current state-of-the-art regularizer-based and guidance methods, showing that our method offers a better diversity-fidelity tradeoff.
- Clarified that our method is not dependent on any particular covariance approximation, and clarified that the learnable scaler need not have any relation to guidance covariance terms.

# Limited Experimental section

There were concerns about the limited span of the experiments, as well as concerns about comparisons to previous guidance-based methods. We have:

- Expanded the experimental section with the Helmholtz Equation and added observation noise to the wave number (with standard deviation up to 50% of the true value).
- Added results from using previous state-of-the-art guidance methods, at the same time clarifying that **our method is more comparable to previous training-time methods**.
- Empirically showed that, by using conflicting objectives, previous guidance-based methods forfeit distributional fidelity by using imbalanced observation models.

We thank the reviewers for their time during this rebuttal process. We believe that through the review process, we have strengthened the paper through writing, additional theoretical results, and additional experiments and measurements. We hope that this reflects in the final presentation of the paper, and hope to hear additional feedback from the Area Chair.

---

### Meta-Review · Area_Chair_McMM · 2025-12-30

**Summary:**

In the initial review, all three reviewers evaluated the paper as below the acceptance threshold (score 2,2,4). The main points raised had to do with: (1) limited novelty, reliance on approximations, limited interpretability, and lack of analysis of the precise effect of the proposed modification to the denoiser (iiNj, N3A7, V5bD); (2) weak experimental verification (iiNj, N3A7, V5bD); (3) lack of clarity regarding the settings in which the proposed method should be chosen over existing methods, like PIDM and PINN (V5bD).

In their response, the authors clarified the main contribution, which seems to have been misunderstood by the reviewers. They further clarified that the approach only introduces an inductive bias to the denoiser, but that the framework remains that of a standard diffusion model and hence the theoretical guarantees of diffusion models apply. Reviewer V5bD engaged in several rounds of discussion, after which the authors claimed that the reviewer responded that they would raise the score from 4 to 6. The AC has no access to this response from the reviewer, and thus cannot verify this.

The AC finds the authors’ clarification about the contribution satisfactory. However, the AC finds that the paper does not provide a satisfactory mathematical definition of the task that the method attempts to solve (namely, what should be considered a good outcome when tilting the distribution of a generative model according to soft constraints). The AC believes that this is part of the confusion of the reviewers, and finds that this lack of clarity is not resolved in the updated manuscript. For example, it is stated in the paper that for a data density function $p(x)$ and a constraint function $c(x)\in\\left\\{0,1\\right\\}$, it is desired to modify the density into $p(x)c(x)$. First, the expression $p(x)c(x)$ is not a density function because it does not integrate to 1. But even if it is normalized, this means that the desired outcome is a model that generates samples only from the intersection of the supports of $p(x)$ and $c(x)$. So, for example in Fig. 1, we would expect the method to not generate samples on the dent, but rather only on the rest of the circle (the dent is outside the intersection between the data distribution and the constraint). This is what PIDM attempts to do (with limited success), but not what the proposed method achieves. Yet, the paper regards the PIDM result as a failure and the SCD result as a success. Therefore, it is not clear what is the mathematical definition of success in this “soft constraint” task. In other words, what density does the method aim for if not $p(x)c(x)$?

In light of the lack of clarity, the AC views the paper as not ready for publication in this ICLR. The AC encourages the authors to improve the mathematical clarity and submit to a future venue.

**Reviewer Concerns:**

The rebuttal provided point-by-point answers to all the reviewers' questions. However, the AC believes that part of the lack of clarity that caused the reviewers to give negative scores still remains, as mentioned above.

**Reviewer Scores:**

**iiNj: score 4.**

The original score was 2. Clarifications about the contribution were provided and experiments were added, hence the increase in score. However, the clarity issue mentioned above remains, so the reviewer would likely not have provided a positive score.

**N3A7: score 4.**

The original score was 2. Clarifications about the contribution were provided and experiments were added, hence the increase in score. The answer about the theoretical guarantee of consistency, which is the same as a standard diffusion model, is not satisfactory. The lack of clarity is not regarding what happens in the case of infinite training data, where the denoiser ignores the constraint and converges to the data distribution. The lack of clarity is regarding what happens in the case of limited training data, so that the inductive bias kicks in. This is the setting in all the illustrations in the paper, where the model does not learn the data distribution but rather a distribution that is also affected by the constraint. The reviewer’s question about a theoretical guarantee relates to those practical settings - what is the distribution learned in such cases? This lack of clarity relates again to the fact that there is no definition of the goal - namely what learned distribution would be considered a success? For this lack of clarity, the AC predicts that the reviewer would not have given a positive score.

**V5bD: score 6.**

The original score was 4. After a discussion, the authors claimed the reviewer promised to increase the score to 6. Although the reviewer comment where this was allegedly stated is not available, the AC views this as reasonable.

---

### Decision · Program_Chairs · 2026-01-26

Reject